# Evolutionary shaping of human brain dynamics

**James C Pang[1,2]\*, James K Rilling[3,4,5], James A Roberts[2†],
Martijn P van den Heuvel[6,7†], Luca Cocchi[2]\*†**

[1]The Turner Institute for Brain and Mental Health, School of Psychological Sciences,
and Monash Biomedical Imaging, Monash University, Victoria, Australia; [2]QIMR
Berghofer Medical Research Institute, Queensland, Australia; [3]Department of
Anthropology, Emory University, Atlanta, United States; [4]Department of Psychiatry
and Behavioral Sciences, Emory University, Atlanta, United States; [5]Yerkes National
Primate Research Center, Emory University, Atlanta, United States; [6]Department of
Complex Traits Genetics, Center for Neurogenetics and Cognitive Research, Vrije
Universiteit Amsterdam, Amsterdam, Netherlands; [7]Department of Clinical Genetics,
Amsterdam UMC, Vrije Universiteit Amsterdam, Amsterdam, Netherlands

**\*For correspondence:**
James.Pang1@monash.edu
(JCP);
Luca.Cocchi@qimrberghofer.edu.
au (LC)

†These authors contributed
equally to this work

**Reviewing Editor:** Claus
Hilgetag, University Medical
Center Hamburg-Eppendorf,
Germany

**Abstract** The human brain is distinct from those of other species in terms of size, organization,
and connectivity. How do structural evolutionary differences drive patterns of neural activity enabling
brain function? Here, we combine brain imaging and biophysical modeling to show that the anatom-
ical wiring of the human brain distinctly shapes neural dynamics. This shaping is characterized by a
narrower distribution of dynamic ranges across brain regions compared with that of chimpanzees,
our closest living primate relatives. We find that such a narrow dynamic range distribution supports
faster integration between regions, particularly in transmodal systems. Conversely, a broad dynamic
range distribution as seen in chimpanzees facilitates brain processes relying more on neural interac-
tions within specialized local brain systems. These findings suggest that human brain dynamics have
evolved to foster rapid associative processes in service of complex cognitive functions and behavior.

## Editor's evaluation

Your intriguing and original study investigates how the characteristic architecture of human brain
networks leads to specific features of global neural dynamics. Your paper addresses a question that
is of wide interest and provides a significant advance in understanding how connectomic features
underlie aspects of the neural dynamics of human versus non-human (chimpanzee) brains. Moreover,
the present approach showcases a powerful computational strategy for identifying structural factors
that may help explain specific cognitive abilities of humans.

## Introduction

An important and unresolved problem in neuroscience is how connectivity, between neurons and
macroscopic brain regions, can give rise to the complex dynamics that underlie behavior and advanced
cognitive functions (*Seyfarth and Cheney, 2014*). Identifying special features of the human brain that
have evolved to support these complex neural dynamics is key in tackling this open question.

It is known that the human brain is approximately three times larger than would be expected in
a primate with the same body mass (*Rilling, 2006*; *Rilling, 2014*). Beyond general growth, neuroim-
aging analyses via magnetic resonance imaging (MRI) have indicated that a greater proportion of the
human brain's cortical surface is allocated to higher-order association cortices compared to primary

sensory and motor areas (*Smaers et al., 2017*; *Avants et al., 2006*; *Van Essen and Dierker, 2007*). This expansion of association areas is accompanied by increased anatomical connectivity (*Ardesch et al., 2019*), providing a structural substrate assumed to enable efficient region-to-region communication and integration of remote neural processes. Studies of the brain's structural wiring, known as the *human connectome*, have shown widespread overlapping topological properties (e.g., small-world and modularity properties; *Ardesch et al., 2019*) with those of other primates (like macaque and chimpanzees), accompanied by subtle but potentially consequential species differences (*van den Heuvel et al., 2016*).

Here, we ask how the abovementioned structural changes shape whole-brain patterns of neural activity supporting brain function. To address this knowledge gap, we combine MRI data with advanced biophysical modeling to generate neural dynamics supported by the human connectome and the connectome of one of our closest living primate relatives: the chimpanzee. The use of biophysical models is crucial to tease apart and explain the neural basis of inter-species differences in whole-brain function, which cannot be achieved with current neuroimaging techniques (*Breakspear, 2017*). By combining this innovative approach with a unique cross-species dataset, we reveal core neural principles likely to explain differences in brain function between humans and non-human primates.

## Results

### Human and chimpanzee connectomes

We begin by creating the connectomes of humans and chimpanzees. We use unique diffusion MRI data for adult humans (*Homo sapiens*) and sex-matched and age-equivalent chimpanzees (*Pan troglodytes*) to reconstruct the connectomes (*Ardesch et al., 2019*; *van den Heuvel et al., 2019*). The connectomes represent cortico-cortical structural connections between 114 species-matched regions in both hemispheres (*Supplementary file 1*) from which we create group-averaged weighted human and chimpanzee connectomes (*van den Heuvel et al., 2019*; *Figure 1A and B*). We then normalize the group-averaged connectomes with respect to their maximum weights. Using the resulting connectomes, we examine connections present in one species but absent in the other (labeled as human-specific and chimpanzee-specific connections; *Figure 1C*). We note that the use of the term 'specific' does not necessarily imply that said connections are unique to each of the species; that is, they are only specific based on comparison of the connectivity strength of connections between the two species in our dataset. We find that intrahemispheric pathways comprise 82.6% (19 out of 23) of human-specific connections and 50% (3 out of 6) of chimpanzee-specific connections, a finding consistent with previous comparative connectome investigations (*Ardesch et al., 2019*; *van den Heuvel et al., 2019*). We also examine the set of connections that are present in both species, termed shared connections (*Figure 1C*), and confirm that there is a strong correlation between connectivity strengths across both species (*Figure 1D*), consistent with previous studies (*Ardesch et al., 2019*; *van den Heuvel et al., 2019*). At the whole-brain level, the human and chimpanzee connectomes largely overlap in their topological organization. In particular, the connectomes show similar levels of small-worldness (small-world propensity [*Muldoon et al., 2016*] values of 0.83 and 0.84 in human and chimpanzee, respectively) and modularity (modularity values of 0.54 and 0.56 in human and chimpanzee, respectively) (*Bullmore and Sporns, 2009*; *Rubinov and Sporns, 2010*). At the regional level, the connectomes exhibit similar distributions of clustering coefficients (*Figure 1E*). On the other hand, human brain regions have significantly shorter path lengths compared to chimpanzee brain regions (*Figure 1F*).

### Modeling neural dynamics

Next, we combine the connectomic data (*Figure 2A*) with a biophysical (generative) model (*Wang, 2002*; *Wong and Wang, 2006*; *Deco et al., 2013*; *Wang et al., 2019*; *Figure 2B*; see Materials and methods) to generate regional synaptic response $S$ across time (*neural dynamics*) specific to each species. The variable $S$ represents the fraction of activated NMDA channels; hence, higher $S$ values correspond to higher neural activity and firing rates. This model has been shown to reproduce empirical human functional neuroimaging data (*Deco et al., 2013*; *Wang et al., 2019*), which we confirmed (*Figure 2—figure supplement 1*). Notably, we also confirmed the model's suitability to match non-human primate data (*Figure 2—figure supplement 1*). These validations of the model on human and

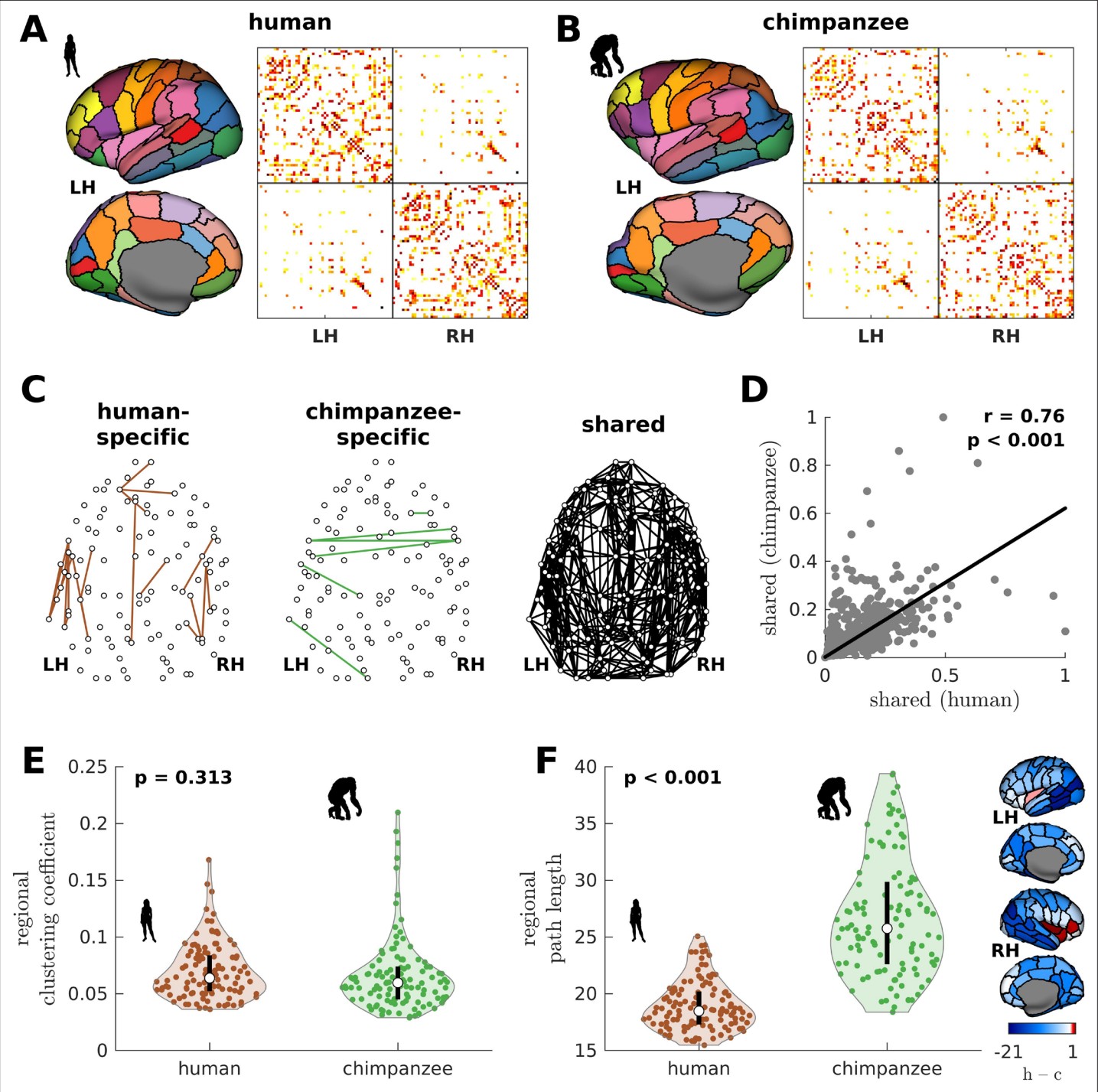

**Figure 1.** Human and chimpanzee connectome properties. (**A, B**) Parcellation and connectome. The surface plots show the 114-region atlas (***Supplementary file 1***) on inflated cortical surfaces. The matrices represent the group-averaged structural connectivity between brain regions. (**C**) Structural connections that are human-specific, chimpanzee-specific, and shared between humans and chimpanzees. (**D**) Association of the weights of the connections shared between humans and chimpanzees. The solid line represents a linear fit with Pearson's correlation coefficient (r) and p value (p). (**E**) Violin plot of the distribution of regional clustering coefficients. Each violin shows the first to third quartile range (black line), median (white circle), raw data (dots), and kernel density estimate (outline). p is the p value of the difference in the mean of the distribution between the species (two-sample t-test). (**F**) Violin plot of the distribution of regional path lengths. Violin plot details are similar to those in **E**. The surface plots show the spatial organization of the difference in path length between the species (i.e., human – chimpanzee) visualized on inflated human cortical surfaces. The negative–zero–positive values are colored as blue–white–red.

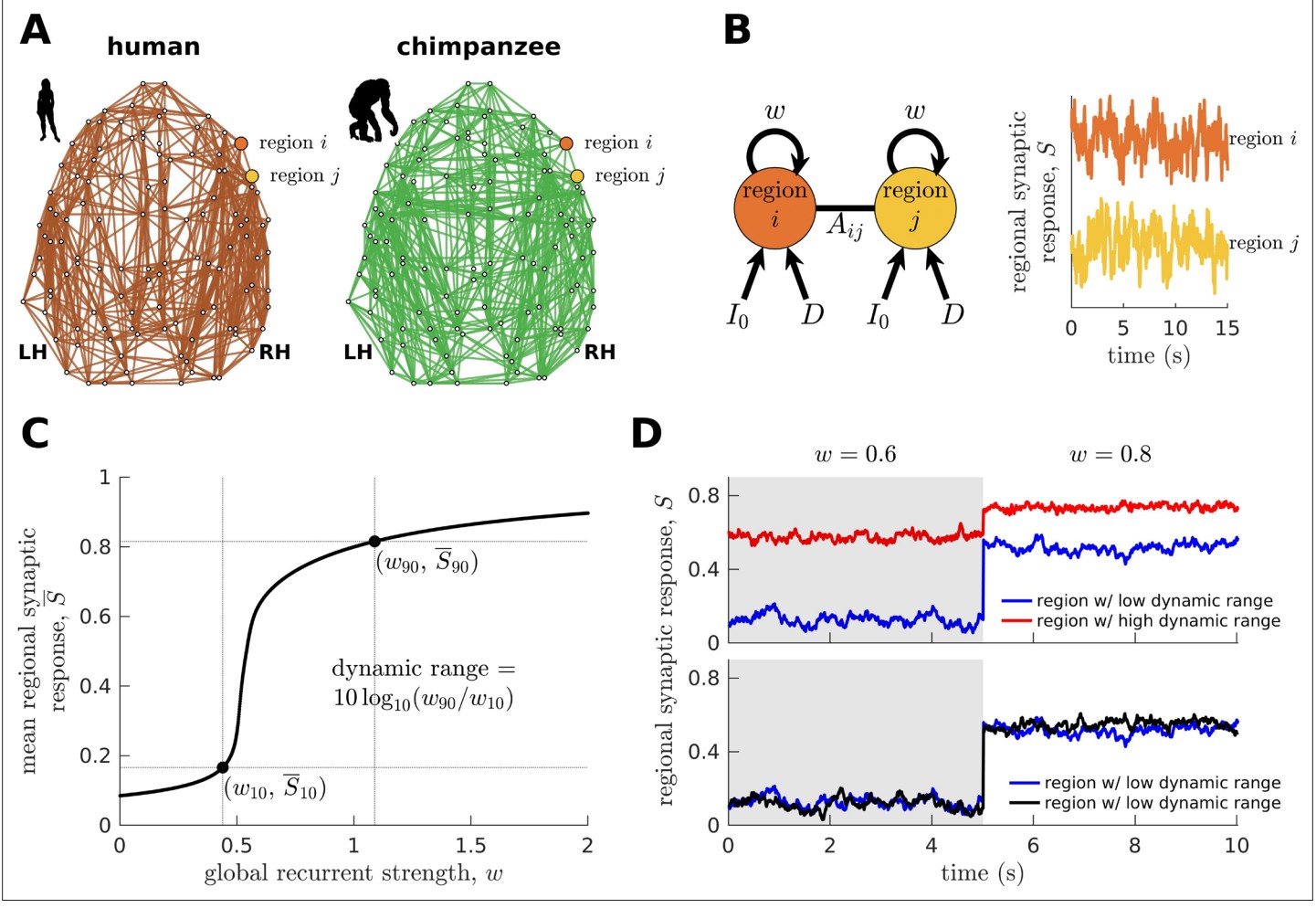

**Figure 2.** Brain network modeling. (**A**) Group-averaged human and chimpanzee networks visualized on the same brain template. Top 20% of connections by strength are shown. (**B**) Schematic diagram of the model. Each brain region is recurrently connected with strength $w$ and driven by an excitatory input $I_0$ and white noise with standard deviation $D$. The connection between regions $i$ and $j$ is weighted by $A_{ij}$ based on the connectomic data. The regional neural dynamics are represented by the synaptic response variable $S$; high $S$ translates to high neural activity. (**C**) Method for calculating the dynamic range of each brain region from its mean synaptic response $\bar{S}$ versus global recurrent strength $w$ curve. Note that $\bar{S}_x = \bar{S}_{min} + (x/100) * (\bar{S}_{max} - \bar{S}_{min})$, with $w_x$ being the corresponding global recurrent strength at $\bar{S}_x$ and $x = \{10, 90\}$. (**D**) Example time series of regions with different (top panel) and similar (bottom panel) dynamic ranges at $w = 0.6$ and 0.8. The time series in the top panel have correlation values (Pearson's r) of 0.06 and 0.08 at $w = 0.6$ and $w = 0.8$, respectively. The time series in the bottom panel have correlations of 0.40 and 0.14 at $w = 0.6$ and $w = 0.8$, respectively.

The online version of this article includes the following figure supplement(s) for figure 2:

**Figure supplement 1.** Validation of simulated dynamics on empirical functional neuroimaging data.

non-human primate data are important to ensure that the outcomes of the model capture meaningful properties of brain activity.

To understand how whole-brain activity patterns emerge, we analyze the intrinsic characteristics of regional neural dynamics. In particular, we determine a brain region's response function, describing the up- or downregulation of its mean activity following global (brain-wide) modulations in the strength of recurrent connections (**Figure 2C**). This process can be linked to how structures in the ascending neuromodulatory systems (e.g., noradrenergic) facilitate the reorganization of cortex-wide dynamics by allowing coordinated communication between otherwise segregated systems (**Shine et al., 2016**; **Shine, 2019**; **Wainstein et al., 2022**). In particular, previous work has shown that neuromodulatory agents can modify the biophysical properties of neurons through various cellular mechanisms (**Shine et al., 2021**). One mechanism is via activation of metabotropic receptors that bring the resting membrane potential of neurons closer to their firing threshold (**Leenders and Sheng, 2005**). This

mechanism can mediate changes in the excitability of brain regions at the subsecond timescale (*Bang et al., 2020*), effectively driving modulations in the regional strength of recurrent connections (i.e., our model's *w* parameter). However, we clarify that the neuromodulation mechanism described above is only one example of many potential mechanisms that can drive changes in regional excitability.

We characterize the shape of the response function (i.e., the slope) demonstrated in *Figure 2C* in terms of the *neural dynamic range*, such that high dynamic range means that a region can respond to a wide range of changes in excitability (*w*), albeit the transition between activity levels is slow (red curve in the top panel of *Figure 2D*). Conversely, regions with a low dynamic range can quickly transition to high levels of activity with small changes in excitability, specifically at a critical intermediate regime (blue curve in the top panel of *Figure 2D*). When the dynamic range is very close to zero, the response function in *Figure 2C* becomes like a step function with infinite slope; hence, the response jumps between low and high activity levels, analogous to a phase transition. Note that our dynamic range is based on an excitability-output function (*Figure 2C*) rather than an input stimulus-output function commonly used in previous studies (*Kinouchi and Copelli, 2006*). Moreover, our definition of dynamic range is different from other definitions based on temporal deviations of a signal with respect to its mean (*Shafiei et al., 2020*). Brain regions with similarly low or high dynamic ranges are more likely to reach equivalent functional states. This can be observed in the example time series at the bottom panel of *Figure 2D*, where regions with similarly low dynamic ranges have activity amplitudes fluctuating at similar levels across varying excitability regimes. Note that similar observations occur for regions with similarly high dynamic ranges. Moreover, regions with similar dynamic ranges have higher levels of correlated activity compared to regions with different dynamic ranges, suggesting better integration (e.g., see correlations of the time series in the bottom versus top panels of *Figure 2D* at corresponding *w* values). Using the neural dynamic range property, we aim to reveal key principles of whole-brain neural dynamics setting humans apart from other species.

## Human brains have more constrained neural dynamics than chimpanzee brains

We find that the response functions of human brain regions (reflecting how activity changes vs. modulations in global excitability) are more similar to one another compared to those of chimpanzees (*Figure 3A*). We quantitatively test this observation by calculating the distribution of dynamic ranges across regions (*Figure 3B*). The results show that the human brain has neural dynamic ranges characterized by a narrower distribution (standard deviation $\sigma$=0.12) as compared to the chimpanzee brain ($\sigma$=0.48). This finding is robust against differences in individual-specific connectomes (*Figure 3—figure supplement 1A,B*), brain volume (*Figure 3—figure supplement 1C*), connection density (*Figure 3—figure supplement 2*), inter-individual variability of connection strengths (*Figure 3—figure supplement 3*), data sample size (*Figure 3—figure supplement 4*), propagation delays between brain regions (*Figure 3—figure supplement 5*), and heterogeneous excitatory input across brain regions (*Figure 3—figure supplement 6*). Moreover, our results are replicated on independent human data from the Human Connectome Project (*Van Essen et al., 2013*; *Figure 3—figure supplement 7*) and a different computational model (*Figure 3—figure supplement 8*).

## Neural dynamic range is spatially organized along the anterior-posterior brain axis

When we map the dynamic ranges onto the anatomical locations of each brain region, we find that both species follow a dominant gradient of neural dynamic ranges spatially organized along the anterior-posterior axis (*Figure 3C*). Specifically, anterior brain regions show neural dynamics with higher dynamic ranges, while posterior regions have lower dynamic ranges. Interestingly, we observe that this dominant gradient is more prominent in chimpanzees than in humans (*Figure 3—figure supplement 9A*). A similar anterior-posterior gradient has also been found in empirical evolutionary expansion maps of the human cortex (*Wei et al., 2019*), with frontal regions being more expanded in humans compared to chimpanzees (*Rilling, 2014*; *Smaers et al., 2017*) and the occipital cortex having relatively similar sizes across the two species (*Figure 3—figure supplement 9B*). Taken together, we additionally observe that highly expanded anterior regions have higher dynamic ranges compared to lowly expanded posterior regions (*Figure 3—figure supplement 9C*).

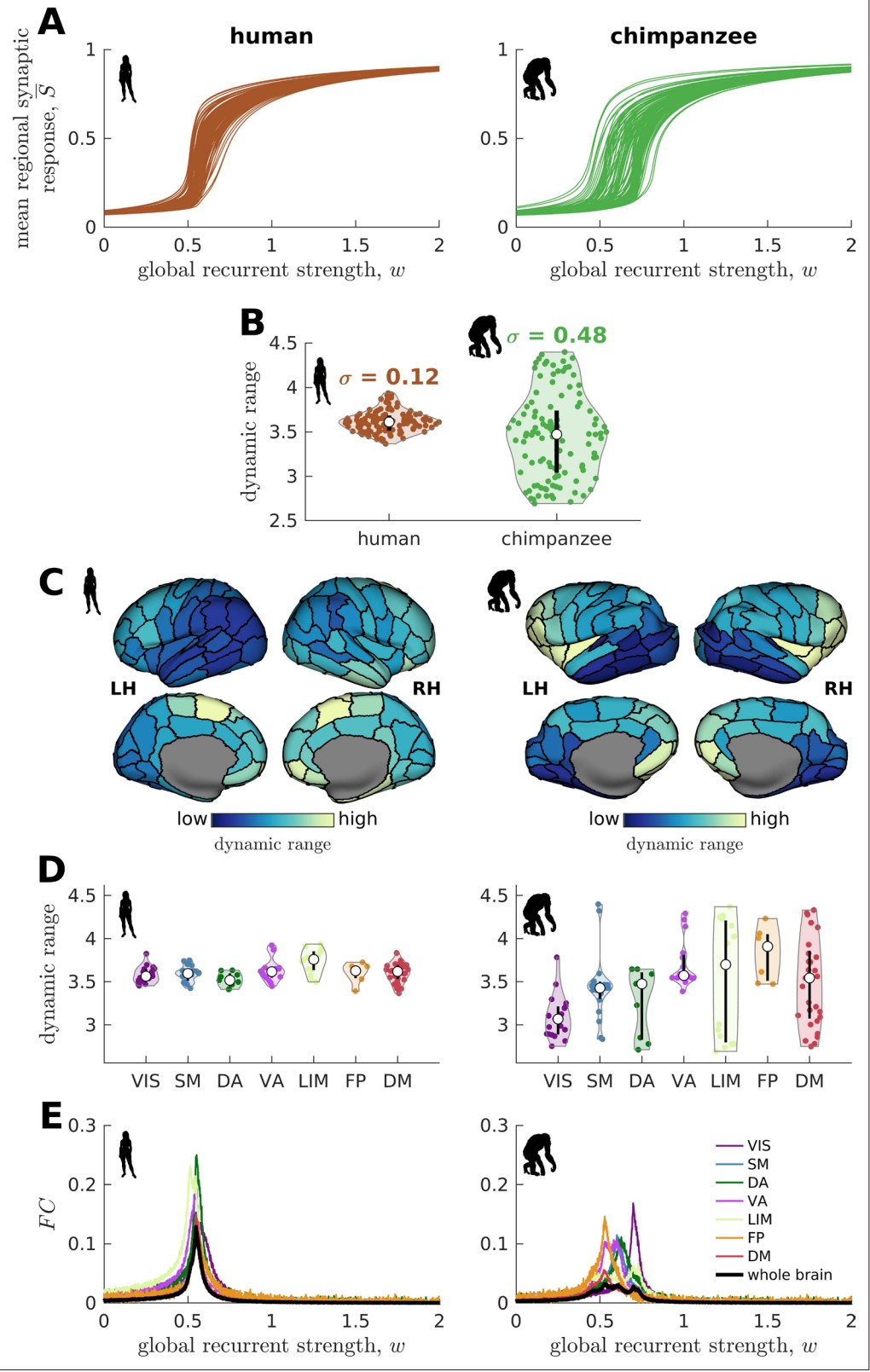

**Figure 3.** Human and chimpanzee neural dynamics. (**A**) Regional neural dynamics as a function of global recurrent strength ($w$). (**B**) Violin plot of the distribution of dynamic ranges across brain regions. Each violin shows the first to third quartile range (black line), median (white circle), raw data (dots), and kernel density estimate (outline). $\sigma$ is the standard deviation of the distribution. (**C**) Spatial organization of dynamic ranges. Data are visualized on

*Figure 3 continued on next page*

*Figure 3 continued*

inflated cortical surfaces. Light color represents high dynamic range and dark color represents low dynamic range. (**D**) Violin plot of the distribution of dynamic ranges in seven canonical brain networks. Violin plot details are similar to those in **B**. (**E**) Simulated average functional connectivity ($FC$) within the networks in **D** as a function of $w$. The black line represents the average $FC$ across the whole brain.

The online version of this article includes the following figure supplement(s) for figure 3:

**Figure supplement 1.** Confirmatory analysis on individual-specific connectomes and accounting for total brain volume.

**Figure supplement 2.** Confirmatory analysis on human and chimpanzee connectomes of equal connection density.

**Figure supplement 3.** Confirmatory analysis accounting for inter-individual variability of connectomic data.

**Figure supplement 4.** Confirmatory analysis on matched sample size.

**Figure supplement 5.** Confirmatory analysis accounting for activity propagation delays between brain regions.

**Figure supplement 6.** Confirmatory analysis accounting for heterogeneous excitatory input across brain regions.

**Figure supplement 7.** Replication of human neural dynamics on an independent dataset.

**Figure supplement 8.** Replication of human and chimpanzee neural dynamics using a different biophysical model (the Wilson-Cowan model).

**Figure supplement 9.** Gradient of dynamic ranges and regional chimpanzee-to-human cortical expansion along the anterior-posterior axis.

**Figure supplement 10.** Anatomical locations of regions clustered according to seven canonical brain networks.

## Similar neural dynamic ranges across regions enables brain network integration

We next ask whether brain regions belonging to specific functional networks have neural dynamics with similar levels of dynamic range. We cluster brain regions into seven common large-scale brain networks according to *Wei et al., 2019*; *Yeo et al., 2011*: Visual (VIS), Somatomotor (SM), Dorsal-Attention (DA), Ventral-Attention (VA, also known as the Salience network), Limbic (LIM), Frontoparietal (FP), and Default-Mode (DM) networks (*Figure 3—figure supplement 10*). These networks represent functionally coupled regions across the cerebral cortex. In humans, brain regions belonging to each functional network show relatively similar neural dynamic ranges (*Figure 3D*). Conversely, in chimpanzees, neural dynamic ranges follow a marked functional hierarchy with cognitive networks (i.e., VA, LIM, FP, and DM) having higher median values than sensory networks (i.e., VIS, SM, and DA). Furthermore, the patterns of within-network changes in functional connectivity (*FC*) versus modulations in global excitability overlap strongly in humans but not in chimpanzees (*Figure 3E*). Thus, similar levels of regional dynamic ranges allow the human brain to better integrate activity within functionally specialized brain networks (colored lines in *Figure 3E*) and the whole brain (black line in *Figure 3E*). This finding is consistent with the higher level of structural integration imposed by the human connectome, as quantified by lower topological path length (*Figure 1F*). Moreover, we find that the heterogeneity in regional path lengths could explain the heterogeneity in neural dynamics, where regions with shorter paths (i.e., lower path length values reflecting higher ability to integrate information between regions) tend to have higher dynamic ranges (*Figure 4*).

## Neural dynamic range differentiates humans and non-human primates

To further test the hypothesis that neural dynamic range is a key feature setting the human brain apart from the brains of other species, we perform similar analyses on other non-human primate connectomic data: macaque (*Macaca mulatta*) (*Shen et al., 2019*) and marmoset (*Callithrix jacchus*) (*Majka et al., 2020*). Neural dynamics are obtained via the model in *Figure 2B* and using weighted connectomes generated from diffusion MRI (for macaques) and invasive tract tracing (for marmosets). The connectomes represent connections between 82 and 55 regions of the macaque and marmoset brains, respectively. Both species have neural response functions more similar to chimpanzees than to humans and with broad dynamic range distributions (*Figure 5*; $\sigma$=0.23 for macaque and $\sigma$=0.31 for marmoset), further validating our results in *Figure 3* across species and across methodological differences in connectome data type and resolution. To verify that the macaque results are not driven by

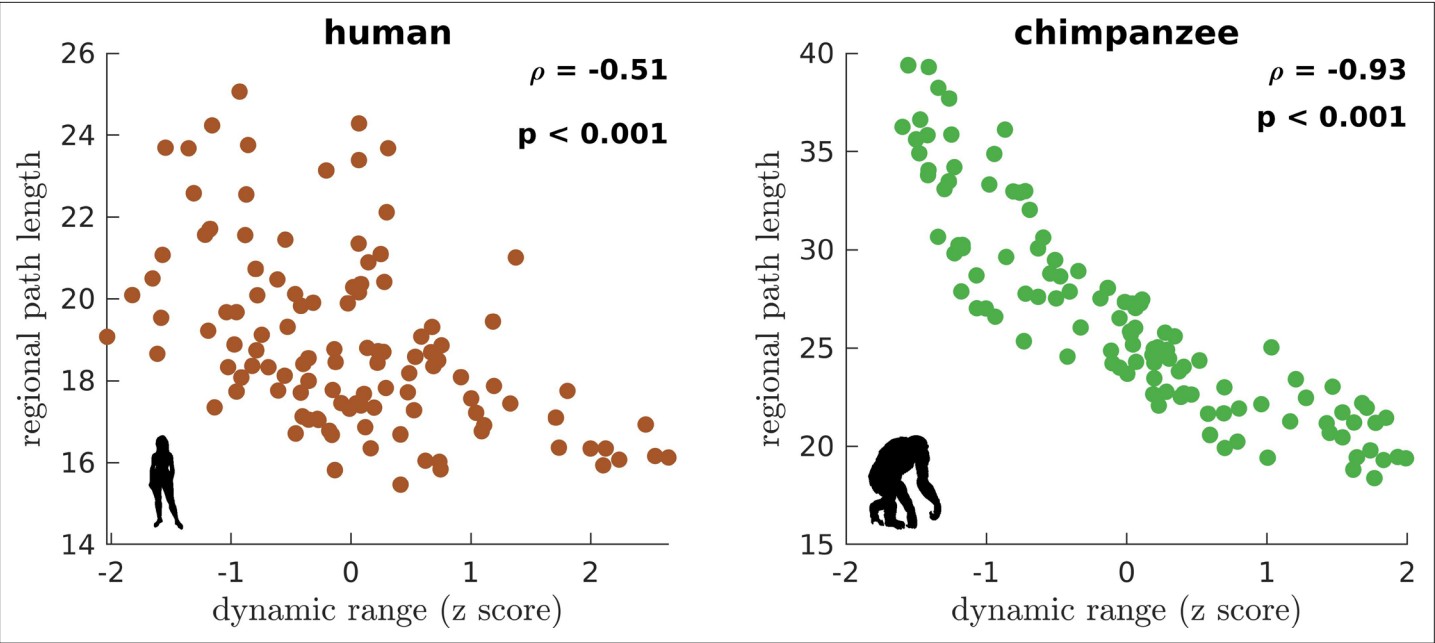

**Figure 4.** Association of the human and chimpanzee connectomes' path length and dynamic range. Average regional path length as a function of z-score-transformed dynamic ranges. $\rho$ is the Spearman rank correlation and p is the p value.

one apparent outlier, as seen in *Figure 5A*, we perform the analysis on independent macaque dataset and find that the results are replicated (*Figure 5—figure supplement 1*).

## Neural dynamic range is linked to the temporal structure of brain activity

To this point, we have shown that the human connectome supports neural dynamics with a narrower distribution of dynamic ranges than the chimpanzee connectome (as well as other non-human primates). However, it remains unclear how dynamic range relates to the temporal structure of neural activity across the brain. Studies have shown that activity within brain regions exhibits a cortex-wide hierarchy of intrinsic neural timescales (*Murray et al., 2014*; *Gao et al., 2020*; *Kiebel et al., 2008*). From these findings, we examine whether a brain region's neural timescale may be related to its neural dynamic range. We extract the timescale by fitting the autocorrelation of the simulated neural activity with a single exponential decay (*Figure 6—figure supplement 1*; see Materials and methods). We find that regional neural timescales (ranges: 0.12–0.24 s for humans and 0.12–0.55 s for chimpanzees) are significantly correlated with dynamic ranges, and this relation is stronger in chimpanzees (*Figure 6A*; this finding also holds for macaques and marmosets as shown in *Figure 6—figure supplement 2A*). This result is consistent with the examples in *Figure 2D*, such that the fast neural timescale of a region with a low dynamic range accommodates the quick transition in response amplitudes of that region when the excitability is increased.

## Neural dynamic range affects the decision-making capacity of human and chimpanzee connectomes

We next ask what would be the implication of the differences in dynamic range distributions between humans and chimpanzees in terms of brain function. We hypothesize that these differences will likely impact the facilitation of whole-brain integration of neural processes, which has been found to be important for performing sensory-perceptual (*Cocchi et al., 2017b*) and complex cognitive tasks (*Shine et al., 2016*) in humans. Note, however, that current whole-brain neuroimaging techniques cannot yet capture the direct effects of neural dynamic range on task performance. Moreover, new chimpanzee brain data via imaging or invasive recordings are not possible to be acquired for ethical reasons.

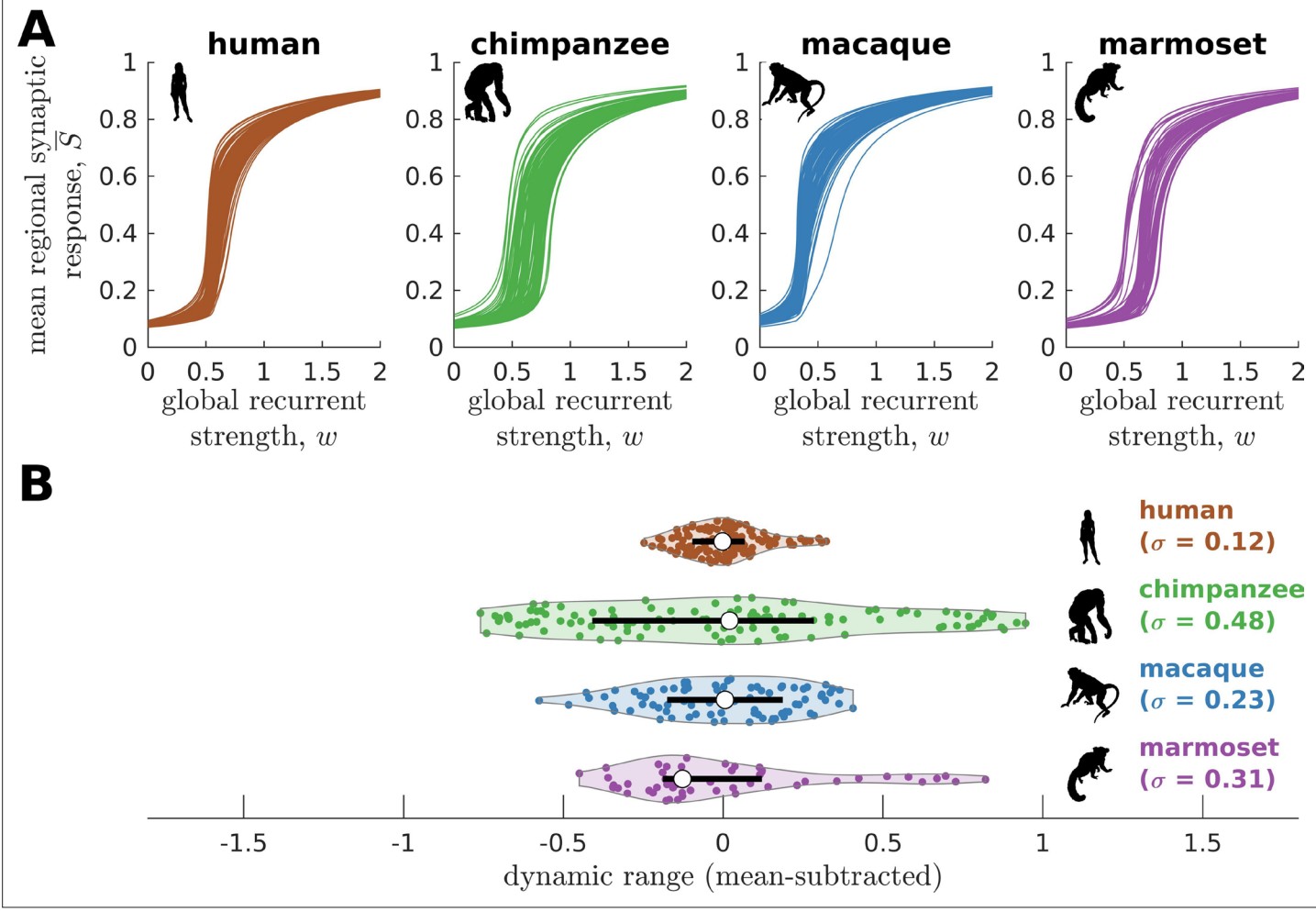

**Figure 5.** Neural dynamics of human and non-human primates. (**A**) Regional neural dynamics as a function of global recurrent strength (*w*) for human, chimpanzee, macaque, and marmoset. (**B**) Violin plot of the distribution of dynamic ranges across brain regions. Each violin shows the first to third quartile range (black line), median (white circle), raw data (dots), and kernel density estimate (outline). The data are mean-subtracted for visual purposes. *σ* is the standard deviation of the distribution.

The online version of this article includes the following figure supplement(s) for figure 5:

**Figure supplement 1.** Replication of macaque neural dynamics on an independent dataset (CoCoMac).

To provide insights into our question, we adopt a computational drift-diffusion model (*Ratcliff et al., 2016*; *Figure 6B*), which is widely used to predict behavioral responses of both humans and animals performing tasks such as decision-making. This model allows us to quantify the capacity of a brain region to achieve a decision threshold by integrating the evidence accumulated by its nearest neighbors. Here, we use the human and chimpanzee connectomes to define a region's neighborhood. The model calculates the accumulation of evidence through time in each brain region via a noise-driven diffusion process until a set threshold is reached (*Figure 6C*; there are two possible thresholds corresponding to a correct or incorrect decision). Then, we estimate each brain region's accuracy in reaching a correct choice across an ensemble of trials (*Figure 6D*) and average these values, representing the likely decision accuracy of the whole brain. Note that we adopt a generalized definition of decision accuracy based on the performance of the connectomes. Specifically, we do not take into account the possibility that only a subset of brain regions could be recruited in the decision-making process. At the end of our simulation, the human brain has a higher accuracy in achieving the correct choice compared to the chimpanzee brain (*Figure 6E*; this finding also holds for macaques and marmosets as shown in *Figure 6—figure supplement 2B*). Interestingly, we find that at earlier times ($t<0.36$ s), the decision accuracy of the human brain is lower than the chimpanzee counterpart.

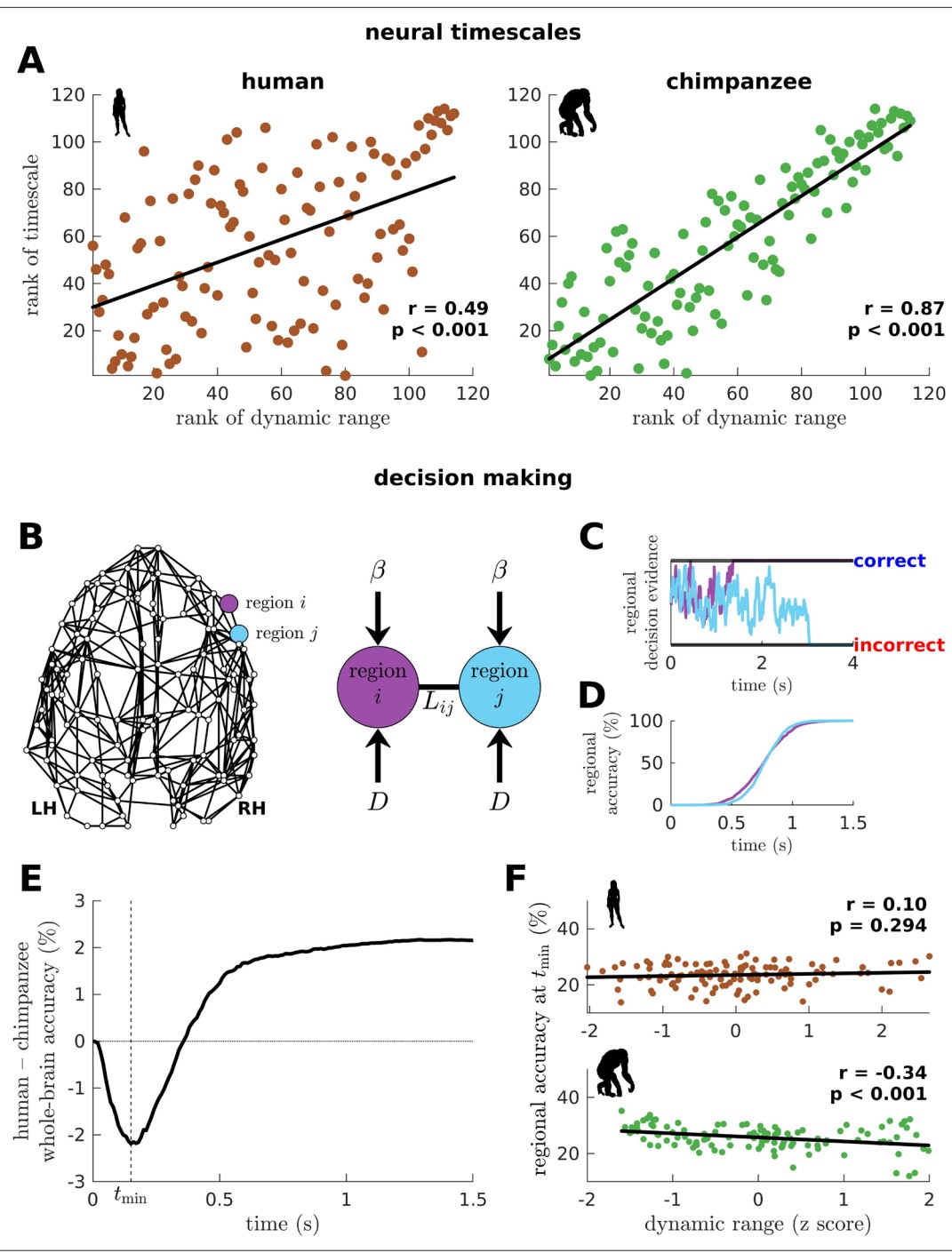

**Figure 6.** Human and chimpanzee neural timescales and connectome decision-making capacity. (**A**) Ranked neural timescales as a function of ranked dynamic ranges. The solid line represents a linear fit with Pearson's correlation coefficient (r) and p value (p). (**B**) Exemplar connectome and schematic diagram of the drift-diffusion model. In the model, each brain region accumulates decision evidence via a diffusion (Brownian) process with drift rate $\beta$ and driving white noise with standard deviation $D$. Regions $i$ and $j$ are connected with Laplacian weight $L_{ij}$ based on the connectomic data. (**C**) Example time series of regional decision evidence across time for regions $i$ and $j$, demonstrating how each region reaches a correct or incorrect decision. (**D**) Regional accuracy curves obtained by simulating the model for an ensemble of trials and calculating the rate of achieving the correct decision. (**E**) Difference in whole-brain accuracy across time between humans and chimpanzees. Whole-brain accuracy represents the average of the accuracy of all regions. The dashed line shows the time ($t_{min}$) at which the difference in accuracy between humans and chimpanzees is most negative (i.e., chimpanzee accuracy>human accuracy).

*Figure 6 continued on next page*

*Figure 6 continued*

(**F**) Regional accuracy at $t_{min}$ (found in **E**) as a function of z-score-transformed dynamic ranges. The solid line represents a linear fit with Pearson's correlation coefficient (r) and p value (p).

The online version of this article includes the following figure supplement(s) for figure 6:

**Figure supplement 1.** Method for calculating neural timescales.

**Figure supplement 2.** Macaque and marmoset neural timescales and their connectome's decision-making capacity.

**Figure supplement 3.** Effects of excitation and inhibition on decision-making capacity of the human and chimpanzee connectomes.

**Figure supplement 4.** Difference in Default-Mode Network (DMN) accuracy across time between humans and chimpanzees.

This finding is driven by regions in the chimpanzee brain with low dynamic ranges that can reach correct decisions quickly (***Figure 6F***).

We next investigate whether levels of excitation and inhibition could have also influenced the difference between the decision accuracy of human and chimpanzee brains at earlier times (***Figure 6E***). Hence, we extend the drift-diffusion model in ***Figure 6B*** by incorporating a self-coupling term parametrized by $\lambda$ (***Figure 6—figure supplement 3A***; $\lambda > 0$ and $\lambda < 0$ corresponds to increased excitation and inhibition, respectively) (***Carland et al., 2015***; ***Lam et al., 2022***). We find that increased excitation leads to faster decision times (***Figure 6—figure supplement 3B***) but poorer overall decision accuracy (***Figure 6—figure supplement 3C***). We also find that increased inhibition extends periods of inferior whole-brain decision accuracy of the human brain compared to the chimpanzee brain at earlier times (***Figure 6—figure supplement 3D***). Interestingly, we also find that the human brain requires the additional level of excitation (i.e., $\lambda = 2.03$) at earlier times in order to reach the level of decision accuracy achieved by the chimpanzee brain (***Figure 6—figure supplement 3E***). This result suggests that our original finding in ***Figure 6E*** could also be driven by higher intrinsic levels of inhibition in the human brain.

## Testing of model predictions on empirical data

We have shown that a brain region's neural dynamic range is tightly linked to the temporal structure of its activity (i.e., neural timescale), suggesting a role of dynamic range in local processing speeds. To test this prediction, we compare empirical cortical T1w:T2w maps, which is an MRI contrast sensitive to myelination, of humans and chimpanzees (***Hayashi et al., 2021***; ***Figure 7A***) to each region's dynamic range. T1w:T2w maps have been shown to be a good macroscale proxy of the cortical processing hierarchy in humans and non-human primates (***Hayashi et al., 2021***; ***Glasser et al., 2014***; ***Burt et al., 2018***), where unimodal sensory-motor regions tend to be highly myelinated and transmodal regions lightly myelinated. Crucially, myelination has been found to be tightly coupled with electrophysiological measures of a region's temporal processing speed (***Gao et al., 2020***). Accordingly, we find that T1w:T2w is inversely related to neural dynamic range (***Figure 7B***). The inverse relation of dynamic range and T1w:T2w is consistent with other studies (***Shafiei et al., 2020***), although their dynamic range metric quantifies the diversity in the fluctuations of activity amplitudes. This result provides an empirical neurobiological support to our findings in ***Figure 6A***, demonstrating that dynamic range is related to a region's processing speed.

We have also shown that the more constrained neural dynamics of the human brain compared to the chimpanzee brain allows better integration of whole-brain activity. We test this prediction on empirical neuroimaging (i.e., functional MRI [fMRI]) data. Because we do not have access to chimpanzee fMRI data, we compare human with macaque data (the same data used in ***Figure 2—figure supplement 1*** to validate our model's suitability). As predicted, we find that $FC$ within large-scale networks is generally higher and more homogeneous in humans compared to macaques (***Figure 8A***). The human brain also has higher network $FC$, as demonstrated by a higher whole-brain $FC$ (p<0.001). Moreover, the human brain has shorter functional paths (i.e., lower path length values) than the macaque brain (***Figure 8B***; similar metric used in ***Figure 1F*** but applied here on $FC$ matrices), which corresponds to better functional integration. We also estimated regional timescales by applying the same technique described in ***Figure 6—figure supplement 1*** on fMRI signals, finding that the human brain has faster

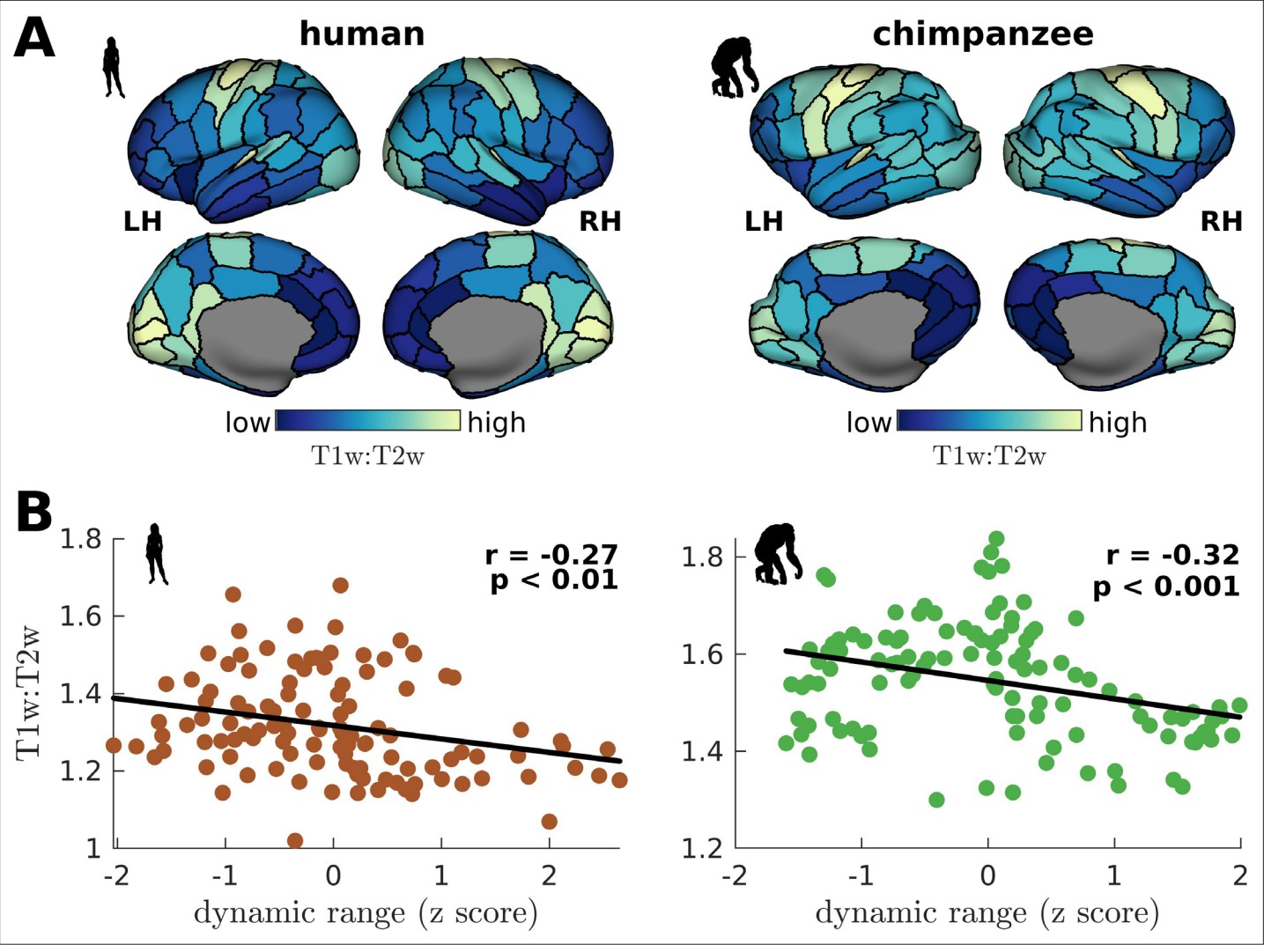

**Figure 7.** Testing of model predictions on T1w:T2w data. (**A**) T1w:T2w maps visualized on inflated cortical surfaces. Light color represents high T1w:T2w value (high myelination) and dark color represents low T1w:T2w value (low myelination). (**B**) Regional T1w:T2w as a function of z-score-transformed dynamic ranges. The solid line represents a linear fit with Pearson's correlation coefficient (r) and p value (p).

timescales than the macaque brain (*Figure 8C*). Overall, we have used available empirical human and non-human primate neuroimaging data to validate two key predictions of our model: neural dynamic range is linked to local processing speed and a narrower dynamic range distribution in humans allows better integration of whole-brain activity.

## Discussion

There is converging evidence that the human brain, as compared to that of our primate relatives, has experienced significant structural reorganizations in association regions throughout evolution (*Rilling, 2014*; *Ponce de León et al., 2021*). Here, we took advantage of a unique neuroimaging dataset of sex-matched and age-equivalent humans and chimpanzees to study how the structural brain wirings of these species support different patterns of whole-brain neural dynamics underpinning brain function. Our results show that these differences determine how the activities of segregated regions are integrated across the brain, giving rise to distinct computational capacities of humans and chimpanzees.

For each species, we determined their brain regions' response functions, which describe the regulation of intrinsic neural activity following brain-wide modulations in excitability. Modulations in global excitability could come from various sources, from external inputs (*Zhang et al., 2020*) to internal

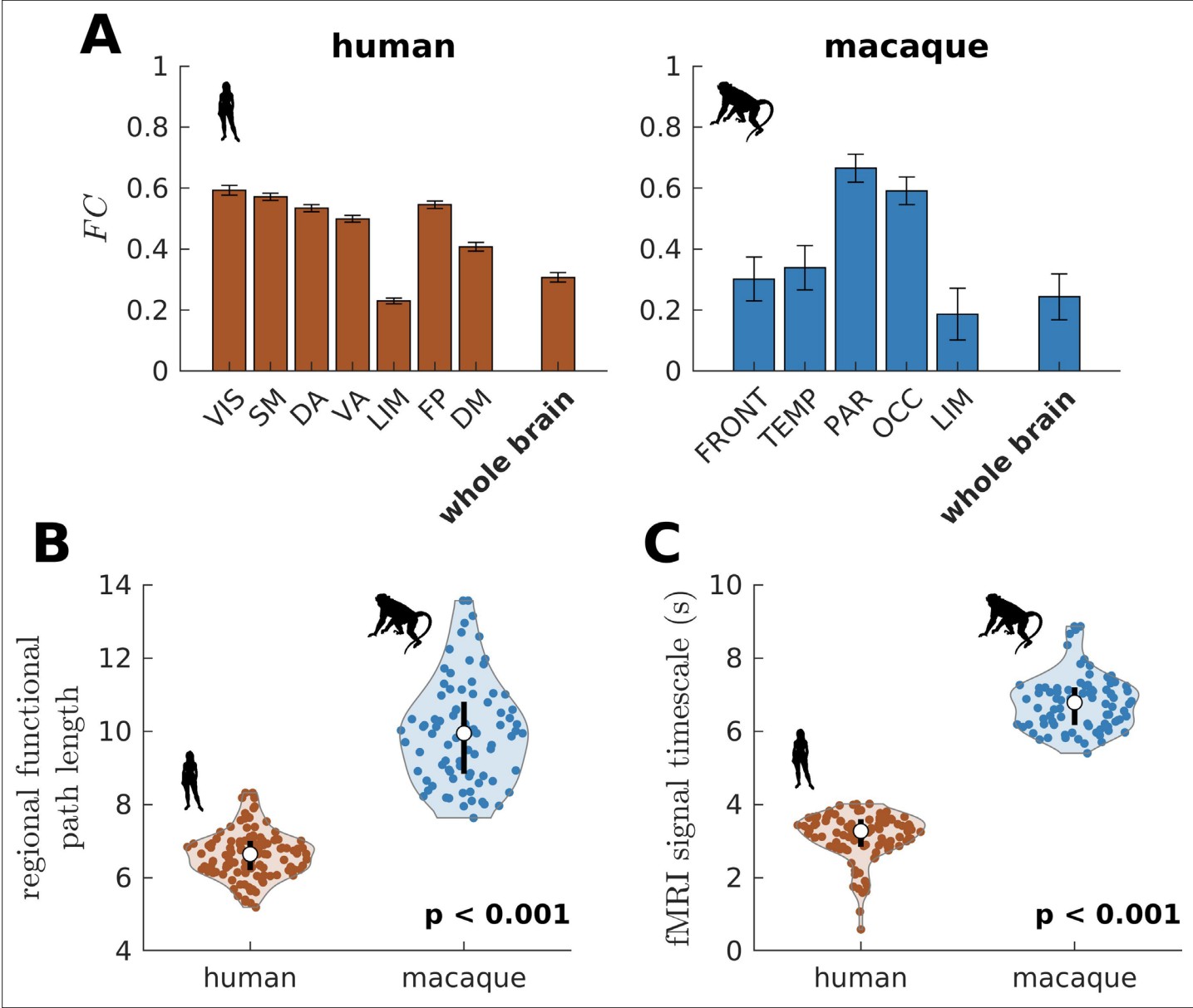

**Figure 8.** Testing of model predictions on functional neuroimaging data. (**A**) Functional connectivity (*FC*) within large-scale networks and across the whole brain of humans and macaques. The human large-scale networks are similar to those defined in *Figure 3D*. The macaque large-scale networks are: FRONT = Frontal; TEMP = Temporal; PAR = Parietal; OCC = Occipital; LIM = Limbic. The error bars are standard errors of the mean across participants (100 humans and 8 macaques). (**B**) Violin plot of the distribution of regional functional path length across brain regions calculated from group-averaged *FC* matrices. Each violin shows the first to third quartile range (black line), median (white circle), raw data (dots), and kernel density estimate (outline). (**C**) Violin plot of the distribution of fMRI signal timescales. Violin plot details are similar to those in **B**. For **B, C**, p is the p value of the difference in the mean of the distribution between the species (two-sample t-test).

influences such as neuromodulation (***Shine, 2019***; ***Wainstein et al., 2022***; ***Bang et al., 2020***). The response function of each region encapsulates all these phenomena, differentiating it from typical input-output response curves (***Kinouchi and Copelli, 2006***; ***Gollo et al., 2016***), with its shape, particularly its slope, characterized by the neural dynamic range. A high dynamic range means that a region can slowly change its activity in response to a wide range of modulations in excitability. Conversely, a low dynamic range means that a region can quickly transition between low and high levels of activity within a narrow range of changes in excitability. We reasoned that the up- or downregulation of local neural activity is crucial to facilitate communication across brain regions, enabling large-scale functional integration and related computations.

In humans, brain regions showed response functions that were more similar to one another than their chimpanzee counterparts. Moreover, the distribution of neural dynamic ranges in humans was narrower than that in chimpanzees and in other non-human primate species (i.e., macaques and marmosets). Thus, neural dynamic range seems to be a unifying feature that sets human brains apart from other primate species. Note, however, that the generalization of the relationship between dynamic range and the evolutionary trajectory of non-human primates is beyond the scope of the current study and remains to be established. These results also propose the hypothesis that relatively subtle evolutionary differences in the connectomes of humans and chimpanzees (*van den Heuvel et al., 2016*) have a marked impact on large-scale neural dynamics supporting the ability of the brain to process information. This could be why human-specific features of connectome organization are vulnerable to brain dysfunction (*van den Heuvel et al., 2019*; *Gollo et al., 2018*).

It has recently been shown that the structural, functional, and genetic properties of cortical regions are spatiotemporally organized in a hierarchical manner (*Kiebel et al., 2008*; *Burt et al., 2018*; *Felleman and Van Essen, 1991*; *Mesulam, 1998*; *Wagstyl et al., 2015*; *Margulies et al., 2016*; *Hasson et al., 2008*). Accordingly, we found that neural dynamic ranges follow a dominant spatial gradient along the anterior-posterior brain axis, with anterior associative regions having higher dynamic ranges than posterior sensory regions. This spatial gradient mirrors the cytoarchitectonic (e.g., cell density) organization observed in the brains of several mammalian species, including rodents and primates (*Charvet et al., 2015*; *Collins et al., 2010*; *Collins et al., 2016*). By analyzing the intrinsic timescales of fluctuations of neural activity (*Murray et al., 2014*; *Gao et al., 2020*; *Kiebel et al., 2008*), we showed that regional neural dynamic ranges correlated with regional neural timescales. Thus, dynamic range seems to capture local information processing speed, providing support to our prior hypothesis that brain regions with similar dynamic ranges are more likely to reach equivalent functional states due to similar processing capacity. At the level of large-scale networks, chimpanzees showed a marked functional hierarchy in neural dynamic ranges compared to humans, with unimodal sensory-perceptual networks having lower values than transmodal associative networks. Importantly, this more pronounced hierarchy in dynamic ranges limits brain network integration in chimpanzees compared to humans. Human brain connectivity appears therefore to have evolved to support neural dynamics that maintain relatively high levels of integration between functionally segregated brain systems (*Ardesch et al., 2019*; *van den Heuvel et al., 2016*).

By using a computational drift-diffusion model (*Ratcliff et al., 2016*), we assessed the functional consequences of neural dynamic range to the human and chimpanzee connectomes' decision-making capacity. Over relatively long processing periods, the human connectome had a higher accuracy in achieving a correct choice compared to the chimpanzee connectome. However, over short periods of processing time, the human connectome performed worse than the chimpanzee counterpart. This latter result was attributed to the (i) more heterogeneous distribution of dynamic ranges in chimpanzees, supporting that diversity in local neural properties is important for rapid computations (*Gollo et al., 2016*), and (ii) higher intrinsic levels of inhibition in the human brain (*Carland et al., 2015*; *Lam et al., 2022*). This regional diversity and intrinsic inhibition may explain why chimpanzees are able to perform at least as well as, or better than, humans in simple sensory-motor tasks (*Martin et al., 2014*). Moreover, our findings provide a possible neural mechanism for why humans generally outperform chimpanzees in tasks requiring longer computational processing (*Tomasello et al., 2005*; *Thornton et al., 2012*). In line with our results, studies have suggested that behavioral differences between humans and chimpanzees are more prominent in complex tasks involving intersubjectivity (e.g., theory of mind; *Herrmann et al., 2007*). This important brain capacity is known to be supported by the activity of the Default-Mode Network (DMN) (*Buckner et al., 2008*; *Spreng et al., 2009*), which displays significant genetic, anatomical, and functional differences between humans and non-human primates (*Wei et al., 2019*; *Bruner et al., 2017*; *Xu et al., 2020*). By making use of our simulations assessing decision-making capacity, we found that the accuracy of DMN for relatively long computations was more accurate in humans compared to chimpanzees (*Figure 6—figure supplement 4*). This finding further suggests that the DMN may be critical in differentiating the functions of human and chimpanzee brains.

Results from the current study suggest that evolution has shaped the human brain to optimize fast transmodal integration of neural activity across brain regions supporting complex functions including social-cultural skills (*Margulies et al., 2016*; *Buckner and Krienen, 2013*). While our results

are consistent with the hypothesis that the human brain has evolved to facilitate rapid associative computations (*Herrmann et al., 2007*), they also highlight that this evolutionary adaptation may hinder rapid processing within functionally specialized systems. The unique properties of human and chimpanzee brain dynamics may therefore be understood as an evolutionary tradeoff between functional segregation and integration. Collectively, our findings inform on the likely neural principles governing evolutionary shifts that could explain the differences in brain function between humans and our closest primate relatives (*Burkart et al., 2017*). Moreover, our work provides a framework to investigate the relationship between whole-brain connectivity and neural dynamics across a wider family of species (e.g., across the mammalian species; *Assaf et al., 2020*) and its effects on complex cognitive processes (e.g., decision-making with more than two alternatives; *Roxin, 2019*).

# Materials and methods
## Connectomic data
### Human and chimpanzee
Diffusion MRI data for 58 humans (*H. sapiens*, 42.5±9.8 years, female) and 22 chimpanzees (*P. troglodytes*, 29.4±12.8 years, female) were taken from previous studies (*Ardesch et al., 2019*; *van den Heuvel et al., 2019*). Procedures were carried out in accordance with protocols approved by the Yerkes National Primate Research Center and the Emory University Institutional Animal Care and Use Committee (YER-2001206). All humans were recruited as healthy volunteers with no known neurological conditions and provided informed consent (IRB00000028). We only provide below a brief account of details of the data and we refer the readers to previous studies for further details. The diffusion MRI acquisition parameters for humans were: spin-echo echo planar imaging (EPI), isotropic voxel size of 2 mm, b-weighting of 1000 s/mm$^2$, 8 $b_0$-scans, and scan time of 20 min. For chimpanzees: spin-echo EPI, isotropic voxel size of 1.8 mm, b-weighting of 1000 s/mm$^2$, 40 $b_0$-scans, and scan time of 60 min. The acquired data were then preprocessed to correct for eddy-current, motion, susceptibility, and head motion distortions. Each participant's cortex was then parcelled using a 114-area subdivision of the Desikan-Killiany atlas (*Desikan et al., 2006*; *Supplementary file 1*; *Figure 1A,B*). Individual undirected connectome matrices were constructed via deterministic tractography to establish cortico-cortical connections between the 114 regions. In line with previous research (*van den Heuvel et al., 2019*), we removed idiosyncratic variations by taking the average weight across individuals of each connection that was consistently found in ≥60% of the individuals, resulting in a group-averaged weighted connectome for each species.

### Macaque
The whole-brain macaque (*M. mulatta*) connectome was derived from eight adult males using diffusion MRI. The data were taken from an open-source dataset, which provides in-depth description of the data (*Shen et al., 2019*). In brief, the diffusion MRI acquisition parameters were: 2D EPI, isotropic voxel size of 1 mm, b-weighting of 1000 s/mm$^2$, 64 directions, and 24 slices. The acquired data were then preprocessed to correct for image distortion and to model fiber directions. Each macaque's cortex was then parcelled into 82 regions following the Regional Map (RM) atlas (*Kötter and Wanke, 2005*). Individual-directed connectome matrices were constructed via probabilistic tractography to establish connections between the 82 regions and thresholded to remove weak connections (thresholds of 0%–35%; see *Shen et al., 2019* for specific optimal threshold values used per individual). A group-averaged weighted connectome was obtained by taking the average weight of non-zero elements of the connectome matrices. It was then thresholded to match the connection density of the group-averaged chimpanzee connectome.

### Marmoset
The marmoset (*C. jacchus*) connectome data were downloaded from the Marmoset Brain Connectivity atlas (marmosetbrain.org) (*Majka et al., 2020*), which is a publicly available repository of cellular resolution cortico-cortical connectivity derived from neuroanatomical tracers. In brief, the connectome was reconstructed from 143 injections of six types of retrograde fluorescent tracers performed on 52 young adult marmosets (1.4–4.6 years, 21 females). Connection weights represent the fraction of labeled neurons found in the target area with respect to the total number of labeled neurons excluding

the neurons in the injected area. The connections were projected onto the Paxinos stereotaxic atlas (*Paxinos et al., 2012*), comprising 116 cortical areas. Individual-directed connectome matrices were constructed by including only areas with pairwise-complete connection values. Thus, the final connectome matrices were 55×55 in size. A group-averaged weighted connectome was obtained by taking the average weight of non-zero elements of the connectome matrices. It was then thresholded to match the connection density of the group-averaged chimpanzee connectome.

## Human HCP

For replication of human results (*Figure 3—figure supplement 7*), minimally preprocessed diffusion MRI data from 100 unrelated healthy young adult participants (29.1±3.7 years, 54 females) were obtained from the Human Connectome Project (HCP) (https://db.humanconnectome.org/; *Van Essen et al., 2013*). In brief, the diffusion MRI acquisition parameters were: isotropic voxel size of 1.25 mm, TR of 5520 ms, TE of 89.5 ms, b-weightings of 1000, 2000, and 3000 s/mm², and 174 slices. The data were then preprocessed for bias-field correction and multi-shell multi-tissue constrained spherical deconvolution to model white matter, gray matter, and cerebrospinal fluid using the MRtrix software (*Tournier et al., 2012*). For each participant, tractograms were generated using 100 million probabilistic streamlines, anatomically constrained tractography (ACT) (*Smith et al., 2012*), the second-order Integration over Fiber Orientation Distributions algorithm (iFOD2), dynamic seeding (*Smith et al., 2015*), backtracking, streamline lengths of 5–250 mm, and spherical-deconvolution informed filtering of tractograms (SIFT2). Each participant's tractogram was projected onto the cortex that was parcellated into 100 regions following the Schaefer atlas (*Schaefer et al., 2018*) to obtain the connectome matrices. A group-averaged weighted connectome was obtained by taking the average weight of non-zero elements of the connectome matrices. It was then thresholded to match the connection density of the chimpanzee connectome.

## Macaque (CoCoMac)

For replication of macaque results (*Figure 5—figure supplement 1*), a directed binary connectome was taken from an open-source dataset (the CoCoMac database) (*Kötter, 2004*; *Honey et al., 2007*). The connectome represents cortico-cortical structural connections between 71 regions derived from histological tract-tracing studies.

## fMRI data

### Human

The empirical human *FC* in *Figure 2—figure supplement 1B* was derived from preprocessed fMRI data of 100 unrelated healthy young adults from HCP (same participants used to calculate the HCP group-averaged connectome above) (*Van Essen et al., 2013*). For each participant, *FC* was calculated by taking pairwise Pearson correlations of the BOLD-fMRI signal across 114 regions (*Supplementary file 1*). A group-averaged *FC* was obtained by taking the average of the individual *FC* matrices.

### Macaque

The empirical macaque *FC* in *Figure 2—figure supplement 1B* was derived from preprocessed fMRI data of eight adult rhesus macaques (same subjects used to calculate the macaque group-averaged connectome above) (*Shen et al., 2019*). For each subject, *FC* was calculated by taking pairwise Pearson correlations of the BOLD-fMRI signal across 82 regions (*Kötter and Wanke, 2005*). A group-averaged *FC* was obtained by taking the average of the individual *FC* matrices.

## Cortical T1w:T2w data

Human and chimpanzee cortical T1w:T2w data, serving as a proxy for myelination, were obtained from *Hayashi et al., 2021*. The T1w:T2w maps were parcellated using our 114-region atlas (*Supplementary file 1*).

## Cortical expansion data

Human and chimpanzee cortical expansion data were obtained from *Wei et al., 2019*. The data represent the ratio of the normalized cortical surface area of each region in the 114-region atlas

(*Supplementary file 1*) between humans and chimpanzees. The normalization was obtained by dividing each region's surface area by the whole cortex's total surface area. Hence, an expansion value greater than 1 means that the relevant region in humans is more expanded compared to the same region in chimpanzees.

### Cortical surfaces

For visualization purposes, we mapped some results into template human and chimpanzee inflated cortical surfaces (*Figure 1A and B*, *Figure 3C*, *Figure 7A*, *Figure 3—figure supplement 9B*, *Figure 3—figure supplement 10*, and *Figure 6—figure supplement 4*) obtained from https://balsa.wustl.edu/study/Klr0B, which is a public repository of data from *Hayashi et al., 2021*.

### Graph theoretical analysis

To investigate the structural property of the connectomes, we leveraged concepts from the field of graph theory (*Muldoon et al., 2016*; *Rubinov and Sporns, 2010*; *Fornito et al., 2016*). In particular, we quantified small-world propensity, modularity, regional clustering coefficient, and regional path length. Small-world propensity quantifies the extent to which the network exhibits a small-world structure (*Muldoon et al., 2016*). Modularity quantifies the extent to which the network may be subdivided into distinct modules (i.e., groups of regions). The clustering coefficient quantifies the probability of finding a connection between the neighbors of a given node (region). Specifically, this metric was estimated by calculating the fraction of triangles around a region. The path length quantifies the level of integration in the network (short path length implying high integration). Path length corresponds to the total topological distance of the shortest path between two regions. For our weighted connectomes, we defined the topological distance to be inversely proportional to the weight of connection (i.e., distance=1/weight). For each brain region, the regional path length was calculated by taking the average of the path lengths between that region and all other regions.

### Computational models

#### Reduced Wong-Wang neural model

To simulate local neural dynamics on the connectome, we used the reduced Wong-Wang biophysical model, also known as the dynamic mean-field model, which is an established model derived from a mean-field spiking neuronal network (*Wang, 2002*; *Wong and Wang, 2006*; *Deco et al., 2013*; *Wang et al., 2019*). Each brain region $i$ is governed by the following nonlinear stochastic differential equation:

$$\dot{S}_i = -\frac{S_i}{\tau_s} + \gamma_s(1 - S_i)H(x_i) + D\nu_i(t), \tag{1}$$

$$H(x_i) = \frac{ax_i - b}{1 - \exp\left[-d\left(ax_i - b\right)\right]} \tag{2}$$

$$x_i = wJS_i + GJ\sum_j A_{ij}S_j + I_0, \tag{3}$$

where $S_i$, $H(x_i)$, and $x_i$ represent the synaptic response variable, firing rate, and total input current, respectively. In the original formulation of the model, the synaptic response variable $S_i$ in *Equation 1* acts as a gating variable that represents the fraction of open (or activated) NMDA channels (*Wang, 2002*; *Wong and Wang, 2006*; *Deco et al., 2013*). Thus, higher values of $S_i$ correspond to higher neural activity. The synaptic response variable is governed by the time constant $\tau_s$ = 0.1 s, saturation rate $\gamma_s$ = 0.641, firing rate $H$, and independent zero-mean Gaussian noise $\nu_i$ with standard deviation $D$ = 0.003. The firing rate $H$ is a nonlinear input-output function defined in *Equation 2* governed by the total input current $x_i$ with constants $a$ = 270 (V nC)$^{-1}$, $b$ = 109 Hz, and $d$ = 0.154 s. The total input current $x_i$ is determined in *Equation 3* by the recurrent connection strength $w$, synaptic coupling $J$ = 0.2609 nA, global scaling constant $G$ = 0.2, connection strength $A_{ij}$ between regions $i$ and $j$, and excitatory subcortical input $I_0$ = 0.33 nA. The parameter values were taken from previous works (*Deco et al., 2013*; *Wang et al., 2019*). Note that the value of the global scaling constant was fixed for all species. This is to ensure that we can directly compare variations in neural dynamics. We simulated the model by numerically solving *Equation 1* using the Euler-Maruyama scheme for a time period of 720 s

and a time step of 0.01 s. We then calculated the time-average value of the synaptic response variable $\bar{S}$ after removing transients, which we used to represent neural dynamics in our analyses.

## Balloon-Windkessel hemodynamic model

To obtain the simulated *FC* in *Figure 3E* and *Figure 2—figure supplement 1*, we fed the neural activity $S_i$ from *Equation 1* to the Balloon-Windkessel hemodynamic model, which is a well-established model for simulating BOLD-fMRI signals (*Stephan et al., 2007*). Note though that this is a simple approximation to more detailed hemodynamic models (*Pang et al., 2017*). Each brain region $i$ is governed by the following nonlinear differential equations:

$$\dot{z}_i = S_i - \kappa z_i - \gamma(f_i - 1), \tag{4}$$

$$\dot{f}_i = z_i, \tag{5}$$

$$\dot{v}_i = \frac{1}{\tau}\left(f_i - v_i^{1/\alpha}\right), \tag{6}$$

$$\dot{q}_i = \frac{1}{\tau}\left\{\frac{f_i}{\rho}\left[1 - (1 - \rho)^{1/f_i}\right] - q_i v_i^{\frac{1}{\alpha} - 1}\right\}, \tag{7}$$

$$\dot{Y}_i = V_0\left[k_1\left(1 - q_i\right) + k_2\left(1 - \frac{q_i}{v_i}\right) + k_3\left(1 - v_i\right)\right], \tag{8}$$

where $z_i$, $f_i$, $v_i$, $q_i$, and $Y_i$ represent the vasodilatory signal, blood inflow, blood volume, deoxyhemoglobin content, and BOLD-fMRI signal, respectively. The model parameters and their values are defined as follows: signal decay rate $\kappa = 0.65$ s$^{-1}$, elimination rate $\gamma = 0.41$ s$^{-1}$, hemodynamic transit time $\tau = 0.98$ s, Grubb's exponent $\alpha = 0.32$, resting oxygen extraction fraction $\rho = 0.34$, resting blood volume fraction $V_0 = 0.02$, and fMRI parameters $k_1 = 4.10$, $k_2 = 0.58$, and $k_3 = 0.53$. The parameter values were taken from previous works (*Stephan et al., 2007*). We simulated the model for a time period of 720 s and the time series were downsampled to a temporal resolution of 0.72 s to match the resolution of typical empirical BOLD-fMRI signals. *FC* was calculated by taking pairwise Pearson correlations of $Y_i$ (after removing transients) across all regions. The within-network *FC* in *Figure 3E* was obtained by taking the average of the *FC* between regions comprising each network.

## Wilson-Cowan neural model

To show that our results generalize beyond our choice of biophysical model, we also simulated local neural dynamics using the Wilson-Cowan model (*Wilson and Cowan, 1972*; *Figure 3—figure supplement 8*). We chose this model because of its known ability to reproduce diverse large-scale neural phenomena (*Papadopoulos et al., 2020*). Each brain region $i$ comprises interacting populations of excitatory ($E$) and inhibitory ($I$) neurons governed by the following nonlinear stochastic differential equations:

$$\dot{S}_i^E = \frac{1}{\tau_E}\left[-S_i^E + \left(1 - S_i^E\right)H_E\left(x_i^E\right) + D_E\nu_i^E(t)\right], \tag{9}$$

$$\dot{S}_i^I = \frac{1}{\tau_I}\left[-S_i^I + \left(1 - S_i^I\right)H_I\left(x_i^I\right) + D_I\nu_i^I(t)\right], \tag{10}$$

$$H_E\left(x_i^E\right) = \frac{1}{1 + \exp\left[-a_E\left(x_i^E - \mu_E\right)\right]}, \tag{11}$$

$$H_I\left(x_i^I\right) = \frac{1}{1 + \exp\left[-a_I\left(x_i^I - \mu_I\right)\right]}, \tag{12}$$

$$x_i^E = w_{EE}S_i^E - w_{EI}S_i^I + G\sum_j A_{ij}S_j^E + G_E P_E, \tag{13}$$

$$x_i^I = w_{IE}S_i^E - w_{II}S_i^I, \tag{14}$$

where $S_i$, $H(x_i)$, and $x_i$ represent the firing rate, non-linear activation function, and weighted sum of firing rates, respectively, for $E$ and $I$ populations. The dynamics of the firing rates $S_i^E$ and $S_i^I$ in *Equations 9; 10* are parameterized by the excitatory time constant $\tau_E = 2.5 \times 10^{-3}$ s, inhibitory time constant $\tau_I = 3.75 \times 10^{-3}$ s, activation functions $H_E$ and $H_I$, and independent zero-mean Gaussian noise $\nu_i^E$ and $\nu_i^I$ with standard deviations $D_E = 5 \times 10^{-5}$ and $D_I = 5 \times 10^{-5}$, respectively. The activation functions

$H_E$ and $H_I$ in **Equations 11; 12** are defined by sigmoids parameterized by the gain constants $a_E = 1.5$ and $a_I = 1.5$ and firing thresholds $\mu_E = 3$ and $\mu_I = 3$. $x_i^E$ and $x_i^I$ are determined in **Equations 13; 14** by the excitatory-excitatory recurrent connection strength $w_{EE} = 16$, excitatory-inhibitory connection strength $w_{EI} = 12$, inhibitory-excitatory connection strength $w_{IE} = 15$, inhibitory-inhibitory recurrent connection strength $w_{II} = 3$, global scaling constant $G = 2$, connection strength $A_{ij}$ between regions $i$ and $j$, and excitatory drive $P_E = 1$ scaled by $G_E = 0.5$. The parameter values were taken from previous works (**Wilson and Cowan, 1972**; **Papadopoulos et al., 2020**). We simulated the model by numerically solving **Equations 9; 10** using the Euler-Maruyama scheme for a time period of 15 s and a time step of 0.001 s. We then calculated the time-average value of the excitatory firing rate $S^E$ after removing transients, which we used to represent neural dynamics in our analyses.

## Drift-diffusion model

To simulate the ability of human and chimpanzee connectomes to reach a binary decision, we implemented a computational drift-diffusion model (**Ratcliff et al., 2016**; **Carland et al., 2015**; **Lam et al., 2022**; **Figure 6B**). Each brain region $i$ is governed by the following stochastic differential equation:

$$\dot{y}_i = \beta_i + \lambda y_i - \sum_j L_{ij} y_j + D\nu_i(t), \tag{15}$$

where $y_i$ is the evidence at time $t$, $\beta_i$ is the drift rate, $\lambda$ is the self-coupling parameter, $L_{ij}$ is the Laplacian weight of the connection between regions $i$ and $j$, and $\nu_i$ is an independent zero-mean Gaussian noise with standard deviation $D$. The Laplacian matrix $L$ is obtained via $L = \mathfrak{D} - A$, where $A$ is the connectivity matrix and $\mathfrak{D}$ is a diagonal matrix of node strengths such that the $i$th diagonal element is $\sum_j A_{ij}$. To focus on the contribution of the connectome itself, we fixed the drift rates of the regions to $\beta_i = 1$ and $D = 1$. We verified that changing these parameter values did not change the results of the study. We also set $\lambda = 0$ in our main results in **Figure 6**, but also extensively varied it in **Figure 6—figure supplement 3** to investigate the effects of excitation (i.e., $\lambda > 0$) and inhibition (i.e., $\lambda < 0$) on the decision-making capacity of the connectomes. Through the simple diffusion process implemented by the model, each region accumulates the decision evidence through time until it reaches a boundary threshold $\theta = \pm 1$ where a decision is said to be reached. Without loss of generality, we assumed $\theta = 1$ to be the correct decision (**Figure 6C**). We simulated the model by numerically solving **Equation 15** using the Euler-Maruyama scheme for a time period of 5 s and a time step of 0.01 s. We then calculated the decision accuracy versus time of each region across an ensemble of 1000 trials (**Figure 6D**).

## Measures of neural dynamics properties

### Neural dynamic range

For each brain region, we analyzed its response function reflecting how mean activity changes versus global modulations in the strength of recurrent connections (i.e., $\bar{S}$ vs. $w$ for the reduced Wong-Wang model and $S^E$ vs. $w_{EE}$ for the Wilson-Cowan model). The response function was characterized in terms of the neural dynamic range, mathematically defined as:

$$\text{dynamic range} = 10 \log_{10} \frac{w_{90}}{w_{10}}, \tag{16}$$

where $w_x$ is the corresponding global recurrent strength at $\bar{S}_x$, with $x = \{10, 90\}$ and

$$\bar{S}_x = \bar{S}_{\min} + \left(\frac{x}{100}\right)\left(\bar{S}_{\max} - \bar{S}_{\min}\right). \tag{17}$$

We then pooled together the dynamic ranges of brain regions for each species of interest to create a distribution with standard deviation $\sigma$.

### Neural timescale

To estimate neural timescales for each brain region, we simulated neural activity via the model described in **Equations 1–3**. We used a global recurrent strength of $w = 0.45$ to produce neural dynamics in a biologically plausible regime; that is, dynamics with relatively low firing rate and not fully synchronized (**Cocchi et al., 2017a**). Following recent studies (**Murray et al., 2014**; **Gao et al., 2020**),

we quantified the neural timescale of each brain region $i$ by fitting the autocorrelation of $S_i(t)$ with a single exponential decay function (via nonlinear least-squares) with form

$$\text{autocorrelation}_i = c_1 e^{-\frac{t}{\tau_i}} + c_2,$$ (18)

where $c_1$ and $c_2$ are fitting constants and $\tau_i$ is the estimated neural timescale (*Figure 6—figure supplement 1*). We verified that fitting with a double exponential decay function did not change the results of the study.

## Additional confirmatory analyses

We performed several confirmatory analyses to check the robustness of our results. In particular, we addressed potential effects of differences in individual-specific connectomes (*Figure 3—figure supplement 1*), differences in connection density (*Figure 3—figure supplement 2*), variability of connection strengths across participants (*Figure 3—figure supplement 3*), differences in human and chimpanzee data sample size (*Figure 3—figure supplement 4*), existence of activity propagation delays between brain regions (*Figure 3—figure supplement 5*), and heterogeneity of excitatory inputs to brain regions (*Figure 3—figure supplement 6*). The procedures for each confirmatory analysis are described below.

### Individual-specific analysis

Brain dynamics resulting from group-averaged connectomes could differ when individual-specific connectomes are used. Hence, we repeated the analysis in *Figure 3* to the connectome of each human and chimpanzee participant to produce *Figure 3—figure supplement 1*.

### Connection density

The resulting connection densities of the group-averaged human and chimpanzee connectomes were different (13.7% vs. 11.6%, respectively). Hence, we created a new human connectome by pruning weak connections such that its density matches the density of the chimpanzee connectome. We then repeated the analysis in *Figure 3* to produce *Figure 3—figure supplement 2*.

### Inter-individual variability of connection strengths

The quality of connectomic data across participants in each species may be different due to potential additional confounds unable to be corrected by the implemented data preprocessing methods. Thus, within each species, we calculated the variability of weights of each connection $A_{ij}$ across participants ($\sigma_{ij}^{\text{human}}$ and $\sigma_{ij}^{\text{chimpanzee}}$). We then rescaled the human and chimpanzee connectomes to match each other's inter-individual variability. Specifically, we multiplied the human connectome weights by $\sigma_{ij}^{\text{chimpanzee}}/\sigma_{ij}^{\text{human}}$ and the chimpanzee connectome weights by $\sigma_{ij}^{\text{human}}/\sigma_{ij}^{\text{chimpanzee}}$. We then repeated the analysis in *Figure 3* to produce *Figure 3—figure supplement 3*.

### Sample sizes

The sample sizes of the human and chimpanzee connectomic data differ (N = 58 for humans and N = 22 for chimpanzees). Hence, we randomly sampled 22 human participants to match the sample size of the chimpanzee data and calculated the corresponding new group-averaged human connectome. The resampling procedure was repeated for 100 trials. For each trial, we repeated the analysis in *Figure 3* to produce *Figure 3—figure supplement 4*.

### Activity propagation delay

The model we used, as described in *Equations 1–3*, assumes that the activity of a brain region propagates and affects instantaneously the activity of all other regions connected to it (i.e., propagation time delay is 0). However, due to the spatial embedding of the brain and the finiteness of activity propagation speeds, we modified the second term of *Equation 3* to incorporate non-zero time delays as follows:

$$x_i(t) = wJS_i(t) + GJ\sum_j A_{ij}S_j\left(t - t_{ij}^d\right) + I_0,$$ (19)

where $t_{ij}^d$ is the propagation time delay between regions $i$ and $j$. We approximated the time delays as $t_{ij}^d = D_{ij}/v$, where $D_{ij}$ is the Euclidean distance between the centroids of regions $i$ and $j$ and $v$ is the propagation speed. We assumed $v$ to be a constant with value 10 m/s (*Deco et al., 2009*) and was the same for both human and chimpanzee brains; choosing other values of $v$ did not change our results. Meanwhile, $D_{ij}$ is specific to human and chimpanzee brains to account for differences in brain sizes. In our analysis, we obtained $D_{ij}$ from one randomly chosen representative human and chimpanzee; the resulting distributions of time delays are shown in *Figure 3—figure supplement 5A*. We then repeated the analysis in *Figure 3* to produce *Figure 3—figure supplement 5B,C*.

## Heterogeneous excitatory input

The model we used, as described in *Equations 1–3*, assumes that the activities of brain regions are driven by a constant excitatory input $I_0$. We tested whether varying $I_0$ per region affects the results of the study. In particular, we incorporated a gradient of excitatory input across the anatomical cortical hierarchy, with unimodal regions having higher inputs and transmodal regions having lower input (*Wang et al., 2019*). Inspired by previous works using a region's total connection strength as a proxy of the cortical hierarchy (*Gollo et al., 2017*; *Pang et al., 2021*), we modified the last term of *Equation 3* as follows:

$$x_i = wJS_i + GJ\sum_j A_{ij}S_j + I_i, \tag{20}$$

$$I_i = I_{\max} - (I_{\max} - I_{\min})\left(\frac{\operatorname{rank}(s_i) - 1}{N - 1}\right), \tag{21}$$

where $I_{\max} = 0.33$ nA, $I_{\min} = 0.28$ nA, $s_i = \sum_j A_{ij}$ is the total connection strength of region $i$, and $N$ is the total number of regions. The resulting excitatory input of regions in human and chimpanzee brains are shown in *Figure 3—figure supplement 6A*. We then repeated the analysis in *Figure 3* to produce *Figure 3—figure supplement 6B,C*. Note that the values of $I_{\max}$ and $I_{\min}$ were chosen to match the spread of estimated excitatory inputs found in *Wang et al., 2019*. However, we verified that using other values of $I_{\max}$ and $I_{\min}$ did not change the results of the study.

## Acknowledgements

The authors thank John Murray, Alex Fornito, and Luke Hearne for valuable scientific discussions. Human HCP data used for replication of results were provided by the Human Connectome Project, Wu-Minn Consortium (Principal Investigators: David Van Essen and Kamil Ugurbil; 1U54MH091657) funded by the 16 NIH Institutes and Centers that support the NIH Blueprint for Neuroscience Research, and by the McDonnell Center for Systems Neuroscience at Washington University. This work was supported by National Health and Medical Research Council grants 1144936 and 1145168 to JAR, Netherlands Organization for Scientific Research grants ALWOP.179, VIDI (452-16-015), and European Research Council Consolidator grant 101001062 to MPVDH, and National Health and Medical Research Council grants 1138711 and 2001283 to LC.

## Additional information

### Funding

| Funder | Grant reference number | Author |
| --- | --- | --- |
| National Health and Medical Research Council | 1144936 | James A Roberts |
| National Health and Medical Research Council | 1145168 | James A Roberts |
| Nederlandse Organisatie voor Wetenschappelijk Onderzoek | ALWOP.179 | Martijn P van den Heuvel |

| Funder | Grant reference number | Author |
|---|---|---|
| Nederlandse Organisatie voor Wetenschappelijk Onderzoek | VIDI (452-16-015) | Martijn P van den Heuvel |
| European Research Council | Consolidator grant 101001062 | Martijn P van den Heuvel |
| National Health and Medical Research Council | 1138711 | Luca Cocchi |
| National Health and Medical Research Council | 2001283 | Luca Cocchi |

The funders had no role in study design, data collection and interpretation, or the decision to submit the work for publication.

#### Author contributions
James C Pang, Conceptualization, Software, Formal analysis, Investigation, Visualization, Methodology, Writing - original draft, Writing – review and editing; James K Rilling, Investigation, Writing – review and editing; James A Roberts, Conceptualization, Resources, Supervision, Funding acquisition, Investigation, Visualization, Methodology, Writing – review and editing; Martijn P van den Heuvel, Conceptualization, Resources, Funding acquisition, Investigation, Visualization, Methodology, Writing – review and editing; Luca Cocchi, Conceptualization, Supervision, Funding acquisition, Investigation, Visualization, Methodology, Writing - original draft, Writing – review and editing

#### Author ORCIDs
James C Pang ⓘ http://orcid.org/0000-0002-2461-2760
Luca Cocchi ⓘ http://orcid.org/0000-0003-3651-2676

#### Ethics
Human subjects: All data were taken from previously published studies and were approved by the respective oversighting ethics committees. Procedures were carried out in accordance with protocols approved by the Yerkes National Primate Research Center and the Emory University Institutional Animal Care and Use Committee (YER-2001206). All humans were recruited as healthy volunteers with no known neurological conditions and provided informed consent (IRB00000028).
Animal experimentation: All data were taken from previously published studies and were approved by the respective oversighting ethics committees. Procedures were carried out in accordance with protocols approved by the Yerkes National Primate Research Center and the Emory University Institutional Animal Care and Use Committee (YER-2001206).

#### Decision letter and Author response
Decision letter https://doi.org/10.7554/eLife.80627.sa1
Author response https://doi.org/10.7554/eLife.80627.sa2

## Additional files

#### Supplementary files
• MDAR checklist
• Supplementary file 1. Table listing the names of 57 cortical regions in each hemisphere.

#### Data availability
All source data and MATLAB codes to perform sample simulations, analyze results, and generate the main and supplementary figures of this study are openly available at https://github.com/jchrispang/evolution-brain-tuning (copy archived at swh:1:rev:ba6ec53aa79bbdc37b54c78e990fae292a5933fc).

The following previously published dataset was used:

| Author(s) | Year | Dataset title | Dataset URL | Database and Identifier |
|---|---|---|---|---|
| Shen K, Bezgin G, Schirner M, Ritter P, Everling S, McIntosh AR | 2019 | A macaque connectome for large-scale network simulations in TheVirtualBrain | https://zenodo.org/record/7011292#.Y0e5UexByS4 | Zenodo, 10.5281/zenodo.7011292 |

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
