## [Editor Report]

Your intriguing and original study investigates how the characteristic architecture of human brain networks leads to specific features of global neural dynamics. Your paper addresses a question that is of wide interest and provides a significant advance in understanding how connectomic features underlie aspects of the neural dynamics of human versus non-human (chimpanzee) brains. Moreover, the present approach showcases a powerful computational strategy for identifying structural factors that may help explain specific cognitive abilities of humans.

---

## [Decision Letter]

**Decision letter after peer review:**

Thank you for submitting your article "Evolutionary shaping of human brain dynamics" for consideration by *eLife*. Your article has been reviewed by two peer reviewers, and the evaluation has been overseen by a Reviewing Editor (Claus Hilgetag) and Christian Büchel as the Senior Editor.

The following individual involved in the review of your submission has agreed to reveal their identity: Bratislav Misic (Reviewer #1).

The reviewers have discussed their reviews with one another, and the Reviewing Editor has drafted this summary to help you prepare a revised submission.

Essential revisions:

1) Both reviewers commented that the observed species differences of simulated brain dynamics bring up the question of the topological differences of the human versus the nonhuman primate connectomes that result in these brain dynamics. Therefore, relevant topological features should be identified, substantiated by additional analyses, and presented more prominently in the paper.

2) Major concerns particularly of Reviewer 2 concern the functional interpretation of the computational modeling results, concretely the inferred dynamic ranges for the nonhuman and human connectomes. Please clarify and substantiate the given functional interpretations, such as the claims regarding facilitated co-activation, functional integration, computational capacity, and decision-making performance, ideally with additional quantitative arguments.

Generally, demonstrate more convincingly, and if possible by quantitative arguments, how the simulated brain dynamics for the human brain confer particular functional advantages. As a test, if one did not know which one was the human connectome, which of the different connectomes would we actually expect to have the best cognitive performance?

*Reviewer #1 (Recommendations for the authors):*

This is an important conceptual advance on the idea that anatomical architecture shapes and constrains neural dynamics. Namely, the authors show that evolutionary differences in connectivity can lead to dynamic differences that support fundamentally different types of communications. These findings are sure to be of wide interest to the field.

The work is rigorous and carried out at a high technical standard: the authors have taken great care to ensure that the connectomes from different species are comparable, that the results hold using multiple methodological approaches, and that they can be replicated in multiple datasets. I also commend the authors for making their code publicly available.

1. "validated biophysical models" These models are certainly realistic and can qualitatively replicate many empirical phenomena, but are they strictly-speaking "validated"?

2. "Specifically, anterior regions (e.g., frontal regions known to be expanded in humans compared to chimpanzees [3,4]) show neural dynamics with higher (more diffuse) dynamic ranges, while posterior regions (e.g., occipital cortex known to be relatively similar in size across the two species) have lower (sharper) dynamic ranges."

Could the authors verify by comparing this map with the evolutionary expansion map from Hill and colleagues?

3. A recent paper from Shafiei and colleagues has investigated empirical patterns of time series features – including several measures of dynamic range – and may be of interest:

Shafiei, G., Markello, R. D., De Wael, R. V., Bernhardt, B. C., Fulcher, B. D., and Misic, B. (2020). Topographic gradients of intrinsic dynamics across the neocortex. *eLife*, 9, e62116.

4. I was surprised that timescale and dynamic range are positively correlated (Figure 4). My intuition would be that time series that have lower autocorrelation fluctuate more rapidly and therefore would assume a greater range of values.

5. I was surprised that the result in Figure S12 was relegated to the supplement. The main finding – that inter-species differences in connectivity give rise to different dynamics – is rather beautiful and interesting, and naturally raises the question of whether any specific topological features of structural connectivity can explain the resulting inter-species differences and regional heterogeneity. The findings are very convincing and could be presented more prominently.

*Reviewer #2 (Recommendations for the authors):*

The paper by Pang et al., demonstrates a promising technique for inferring functionally relevant information from connectome data, which in turn can be used to compare humans with other primates. Such comparisons facilitate the formation of evolutionary hypotheses to account for the particularities of human behavior and cognition.

This paper is a good example of the use of dynamical modeling to distill interpretable information from neuroimaging data. The results are qualitatively clear: the differences between humans and other primates are visible in the figures. The findings related to a gradient of dynamic ranges along the anterior-posterior axis are also of wide interest and can be readily compared with studies on anatomical and physiological gradients. The study involves validation using a second computational model, enhancing the robustness of the results. The connection with computational capacity is also very interesting. Further, the model is used to make specific predictions about functional connectivity, which are verified. The results will be of interest to several subdisciplines within neuroscience: in particular, the link between connectivity and computational capacity has the potential to dovetail with more fine-grained types of data collected in other species, as well as with computational models of decision-making and evidence-accumulation.

The primary weakness of the paper lies in the difficulty of interpreting the broader meaning of the results. Readers may not readily understand what the behavioral and psychological consequences of high and low dynamic ranges are. Some further elaboration of this concept, perhaps with examples from perception and decision-making, will increase the potential readership. Additional explanation of why responses to changes in global recurrent strength are relevant to the local dynamic ranges of each cortical region/network will also be helpful.

The interpretability issue also arises for the treatment of computational capacity: some elaboration of the differences and trade-offs when comparing accuracy in humans and chimpanzees will help with readability. The concept of "decision accuracy of the whole brain" is also somewhat obscure: it is not obvious why a decision should be construed as the result of independent drift-diffusion processes. It may also be helpful to comment briefly on how such a framework can be extended when a decision process is not a two-alternative forced choice.

The nature of the anatomical difference between humans and other primates is also somewhat unclear. The results are described as suggesting evidence for structural changes over the course of evolution. The difference in dynamic range distribution between humans and other primates serves as indirect evidence of these structural changes. But the results do not seem to include a direct comparison of the structural data (prior to modeling, using descriptive statistics) across the various primate species. In addition to this empirical question, the model suggests the possibility of generating specific hypotheses regarding the trajectory of human evolution. Is increased speed/myelination the key factor, or does the overall magnitude of connection weights matter too? What alterations of the chimpanzee connectivity dataset would bring the dynamic range distribution in line with that of humans? The information required to suggest answers to such questions seems to be in the paper, but it is not easy for the reader to piece it together.

Given that the methods come at the end, some comments on terminology would be helpful in the introduction and/or the results. It is not clear why the term "gating" is used instead of "output" or "response". A reader without a computational background may not be familiar with this term.

Why would neuromodulation only affect recurrent strength and not the off-diagonal terms of the connection matrix (matrix A)?

The drift-diffusion model may not be sufficient to account for certain aspects of decision-making, such as urgency (e.g., Carland et al., 2015), or direct inhibition-mediated competition between alternatives. The differences in speed between humans and chimpanzees could conceivably point to differences in the amount and/or timescale of inhibition. Inhibition seems to be higher in humans than in other mammals: this may be relevant to the slower integration of evidence. Even in a model that lumps excitation and inhibition together as positive and negative connection weights, it may be illuminating to know if the E/I ratio sheds light on the speed of arriving at the decision threshold. This is not essential, but inhibition-related inferences would greatly enhance the interest of the paper.

Using the term "diffuse" for high dynamic range seems a bit confusing without further elaboration.

Figure 1. A qualitative explanation of the sigmoidal shape would be useful. How can we interpret the two knee points of the sigmoid, as well as the slope? Is a very steep sigmoid (or a step function) equivalent to an all-or-nothing response to the crossing of a threshold in recurrent activity?

Page 1 Line 13: Some examples of topological properties would be useful.

Page 3 line 10: "Brain regions with similar dynamic ranges are more likely to coactivate, allowing efficient region-to-region integration of neural processes." Why are they more likely to coactivate? Is there a probabilistic argument? One might assume instead that brain regions with high/diffuse dynamic ranges would be more likely to coactivate, since there are more opportunities for them to become active in the first place.

Reference

Carland, M. A., Thura, D., and Cisek, P. (2015). The urgency-gating model can explain the effects of early evidence. Psychonomic Bulletin and Review, 22(6), 1830-1838.

---

## [Author Response]

Essential revisions:1) Both reviewers commented that the observed species differences of simulated brain dynamics bring up the question of the topological differences of the human versus the nonhuman primate connectomes that result in these brain dynamics. Therefore, relevant topological features should be identified, substantiated by additional analyses, and presented more prominently in the paper.

We thank the Reviewing Editor and Reviewers for this suggestion. Accordingly, we have added a new section entitled “Human and chimpanzee connectomes” at the beginning of the Results section to better highlight the structural and topological similarities and differences in the human and chimpanzee connectomes. In this section, we have included new analyses of connections specific to and shared between the two species. We have also provided relevant graph theoretical measures, such as small-worldness, modularity, clustering coefficient, and path length, to better summarize the topological organization of the human and chimpanzee connectomes (new Figure 1). In line with Reviewer 1’s suggestion, we have also added a new Figure 4 to emphasize the relationship between regional path length and dynamic range. As a result, we have added new text Lines 62-79, 206-211, 535-547 and new Figures 1 and 4. Further details about the specific changes can be found in Responses 1.5 and 2.0d.

2) Major concerns particularly of Reviewer 2 concern the functional interpretation of the computational modeling results, concretely the inferred dynamic ranges for the nonhuman and human connectomes. Please clarify and substantiate the given functional interpretations, such as the claims regarding facilitated co-activation, functional integration, computational capacity, and decision-making performance, ideally with additional quantitative arguments.Generally, demonstrate more convincingly, and if possible by quantitative arguments, how the simulated brain dynamics for the human brain confer particular functional advantages. As a test, if one did not know which one was the human connectome, which of the different connectomes would we actually expect to have the best cognitive performance?

In line with Reviewer 2’s suggestions, we have replaced the term “co-activate” with “reach similar functional states” to avoid confusion. We have also expanded the text to better describe: (i) how metabotropic neuromodulation can change brain region excitability; (ii) the meaning of dynamic range and its functional interpretation, especially in the context of integration; and (iii) the synaptic response variable and response function (we demonstrate this via the new Figure 2D). As a result, we have added new text Lines 98-99, 111-137, 252-254, 398, 557-559 and a new panel D in Figure 2. Further details about the specific changes can be found in Responses 1.4, 2.0b, 2.1, 2.2, 2.4, 2.5, and 2.7.

We have also replaced the term “computational capacity” with “decision-making capacity”. This amendment better reflects the nature of our analysis, ensuring that the reader appreciates that we are not considering cellular-level patterns of activity (e.g., neuronal spikes). We have also clarified our term “whole-brain accuracy” as indicating the average accuracy across all regions. In line with Reviewer 2’s suggestions, we have also included a new analysis of the effects of excitation and inhibition on the decision-making capacity of human and chimpanzee connectomes. We have found that inhibition could also explain the poor early decision accuracy emerging from the human connectome compared to that of the chimpanzee. As a result, we have added new text Lines 266-267, 290-293, 299-311, 410-414, 435-438, 634-636 and new Figure 6—figure supplement 3. Further details about the specific changes can be found in Responses 2.0c and 2.3.

Finally, we believe that the question of testing connectomes that give rise to the best cognitive performance is difficult to answer as “best cognitive performance” cannot be easily defined. Humans outperform chimpanzees in several but not necessarily all cognitive domains; for example, chimpanzees outperform humans in some working memory tasks (Inoue and Matsuzawa, 2007, Current Biology). Our work can only provide insights into how this can be answered via a cross-species comparison of connectome properties and (inferred) neural dynamics. The revised manuscript shows that the human connectome is topologically distinct from the chimpanzee connectome, having more intrahemispheric specific connections and shorter regional path lengths, an observation consistent with studies across primates (Ardesch et al., 2022, Cerebral Cortex). Our extensive analyses also show that dynamic range is related to processing speed and that a narrow distribution of dynamic ranges facilitates efficient integration of functionally segregated processes. Further computational analysis demonstrates that these properties endow the human connectome with the capacity to support better decision accuracies over long periods of time. Our results provide an overview of what structural and functional properties of the brain may distinguish humans from chimpanzees.

Reviewer #1 (Recommendations for the authors):This is an important conceptual advance on the idea that anatomical architecture shapes and constrains neural dynamics. Namely, the authors show that evolutionary differences in connectivity can lead to dynamic differences that support fundamentally different types of communications. These findings are sure to be of wide interest to the field.The work is rigorous and carried out at a high technical standard: the authors have taken great care to ensure that the connectomes from different species are comparable, that the results hold using multiple methodological approaches, and that they can be replicated in multiple datasets. I also commend the authors for making their code publicly available.

We thank the Reviewer for their careful evaluation and appreciation of our manuscript.

1. "validated biophysical models" These models are certainly realistic and can qualitatively replicate many empirical phenomena, but are they strictly-speaking "validated"?

We understand the reviewer's concern and have removed the word “validated” in the Introduction and Results sections.

Line 50: … biophysical models …

Line 96: … biophysical (generative) model …

2. "Specifically, anterior regions (e.g., frontal regions known to be expanded in humans compared to chimpanzees [3,4]) show neural dynamics with higher (more diffuse) dynamic ranges, while posterior regions (e.g., occipital cortex known to be relatively similar in size across the two species) have lower (sharper) dynamic ranges."Could the authors verify by comparing this map with the evolutionary expansion map from Hill and colleagues?

We thank the Reviewer for this suggestion; we have now added a new analysis. We used an existing cortical expansion map from Wei et al., (2019, Nature Communications) to investigate variations of regional expansion across the anterior-posterior axis. The expansion map generated by Wei et al., (2019) is more suitable for our work as it is based on comparisons between humans and chimpanzees. On the other hand, the evolutionary expansion map from Hill et al., (2010, PNAS) was obtained via comparisons between humans and macaques.

We have added the following text to the Results and Materials and methods sections and have added a new Figure 3—figure supplement 9:

Lines 183-191: “Specifically, anterior brain regions show neural dynamics with higher dynamic ranges, while posterior regions have lower dynamic ranges. Interestingly, we observe that this dominant gradient is more prominent in chimpanzees than in humans (Figure 3—figure supplement 9A). A similar anterior-posterior gradient has also been found in empirical evolutionary expansion maps of the human cortex [27], with frontal regions being more expanded in humans compared to chimpanzees [3,4] and the occipital cortex having relatively similar sizes across the two species (Figure 3—figure supplement 9B). Taken together, we additionally observe that highly expanded anterior regions have higher dynamic ranges compared to lowly expanded posterior regions (Figure 3—figure supplement 9C).”

Lines 523-528: “Human and chimpanzee cortical expansion data were obtained from [27]. The data represent the ratio of the normalized cortical surface area of each region in the 114-region atlas (Table S1) between humans and chimpanzees. The normalization was obtained by dividing each region’s surface area by the whole cortex’s total surface area. Hence, an expansion value greater than 1 means that the relevant region in humans is more expanded compared to the same region in chimpanzees.”

3. A recent paper from Shafiei and colleagues has investigated empirical patterns of time series features – including several measures of dynamic range – and may be of interest:Shafiei, G., Markello, R. D., De Wael, R. V., Bernhardt, B. C., Fulcher, B. D., and Misic, B. (2020). Topographic gradients of intrinsic dynamics across the neocortex. eLife, 9, e62116.

We thank the reviewer for pointing out this reference. We note, however, that the definition of dynamic range by Shafiei et al., (2020, *eLife*) captures dynamical features at the time-series level, which is different from our definition based on the shape of the excitability-output function.

We have added the following text to the Results section to make the readers aware of differences in the definition of dynamic range in the literature and also relate our findings to those of Shafiei et al., (2020):

Lines 128-130: “Moreover, our definition of dynamic range is different from other definitions based on temporal deviations of a signal with respect to its mean [25].”

Lines 322-324: “The inverse relation of dynamic range and T1w:T2w is consistent with other studies [25], although their dynamic range metric quantifies the diversity in the fluctuations of activity amplitudes.”

4. I was surprised that timescale and dynamic range are positively correlated (Figure 4). My intuition would be that time series that have lower autocorrelation fluctuate more rapidly and therefore would assume a greater range of values.

We first wish to clarify that our dynamic range does not measure the range of values attained by a single time series. Rather, we defined dynamic range as the shape of the excitability-output response function (i.e., related to its slope). Here, a high dynamic range means that the average response can slowly vary across a wide range of changes in excitability. On the other hand, a low dynamic range means that the response can quickly transition from low to high values with small changes in excitability, especially at an intermediate regime. This transition becomes like a jump phase transition the closer the value of the dynamic range is to zero. With this definition, a low dynamic range should correspond to a short (i.e., fast) timescale because rapid fluctuations can accommodate the quick transitions in response amplitudes.

We have added the following text to the Results section to better define dynamic range and better discuss our timescale results:

Lines 119-130: “We characterize the shape of the response function (i.e., the slope) demonstrated in Figure 2C in terms of the *neural dynamic range*, such that high dynamic range means that a region can respond to a wide range of changes in excitability (w), albeit the transition between activity levels is slow (red curve in the top panel of Figure 2D). Conversely, regions with a low dynamic range can quickly transition to high levels of activity with small changes in excitability, specifically at a critical intermediate regime (blue curve in the top panel of Figure 2D). When the dynamic range is very close to zero, the response function in Figure 2C becomes like a step function with infinite slope; hence, the response jumps between low and high activity levels, analogous to a phase transition. Note that our dynamic range is based on an excitability-output function (Figure 2C) rather than an input stimulus-output function commonly used in previous studies [24]. Moreover, our definition of dynamic range is different from other definitions based on temporal deviations of a signal with respect to its mean [25].”

Lines 252-254: “This result is consistent with the examples in Figure 2D, such that the fast neural timescale of a region with a low dynamic range accommodates the quick transition in response amplitudes of that region when the excitability is increased.”

5. I was surprised that the result in Figure S12 was relegated to the supplement. The main finding – that inter-species differences in connectivity give rise to different dynamics – is rather beautiful and interesting, and naturally raises the question of whether any specific topological features of structural connectivity can explain the resulting inter-species differences and regional heterogeneity. The findings are very convincing and could be presented more prominently.

We thank the Reviewer for their enthusiasm and interest in our results. Following their suggestion, we have added new analyses and a figure to the main text. These additions emphasize inter-species differences in the underlying topological features of the connectomes and their relation to observed differences in neural dynamics. In particular, we have added a new Results section “Human and chimpanzee connectomes” to highlight the similarities and differences in the topological features of human and chimpanzee connectomes. In this new section, we have included analyses of anatomical connections that are specific to and shared between the two species. We have also expanded our graph theoretical analysis to include global (small-worldness and modularity) and regional (such as clustering coefficient and path length) topological measures commonly used in the literature.

We have added the following text to the Results and Materials and methods sections and we have removed the old Figure S12. In addition, two new figures have been added to the main text (Figures 1 and 4):

Lines 62-79: “We then normalize the group-averaged connectomes with respect to their maximum weights. Using the resulting connectomes, we examine connections present in one species but absent in the other (labeled as human-specific and chimpanzee-specific connections; Figure 1C). We note that the use of the term “specific” does not necessarily imply that said connections are unique to each of the species; i.e., they are only specific based on comparison of the connectivity strength of connections between the two species in our dataset. We find that intrahemispheric pathways comprise 82.6% (19 out of 23) of human-specific connections and 50% (3 out of 6) of chimpanzee-specific connections, a finding consistent with previous comparative connectome investigations [7,10]. We also examine the set of connections that are present in both species, termed shared connections (Figure 1C), and confirm that there is a strong correlation between connectivity strengths across both species (Figure 1D), consistent with previous studies [7,10]. At the whole-brain level, the human and chimpanzee connectomes largely overlap in their topological organization. In particular, the connectomes show similar levels of small-worldness (small-world propensity [11] values of 0.83 and 0.84 in human and chimpanzee, respectively) and modularity (modularity values of 0.54 and 0.56 in human and chimpanzee, respectively) [12,13]. At the regional level, the connectomes exhibit similar distributions of clustering coefficients (Figure 1E). On the other hand, human brain regions have significantly shorter path lengths compared to chimpanzee brain regions (Figure 1F).”

Lines 206-211: “This finding is consistent with the higher level of structural integration imposed by the human connectome, as quantified by lower topological path length (Figure 1F). Moreover, we find that the heterogeneity in regional path lengths could explain the heterogeneity in neural dynamics, where regions with shorter paths (i.e., lower path length values reflecting higher ability to integrate information between regions) tend to have higher dynamic ranges (Figure 4).”

Lines 535-547: “To investigate the structural property of the connectomes, we leveraged concepts from the field of graph theory [11,13,74,76]. In particular, we quantified small-world propensity, modularity, regional clustering coefficient, and regional path length. Small-world propensity quantifies the extent to which the network exhibits a small-world structure [11]. Modularity quantifies the extent to which the network may be subdivided into distinct modules (i.e., groups of regions). The clustering coefficient quantifies the probability of finding a connection between the neighbors of a given node (region). Specifically, this metric was estimated by calculating the fraction of triangles around a region. The path length quantifies the level of integration in the network (short path length implying high integration). Path length corresponds to the total topological distance of the shortest path between two regions. For our weighted connectomes, we defined the topological distance to be inversely proportional to the weight of connection (i.e., distance = 1/weight). For each brain region, the regional path length was calculated by taking the average of the path lengths between that region and all other regions.”

Reviewer #2 (Recommendations for the authors):The paper by Pang et al., demonstrates a promising technique for inferring functionally relevant information from connectome data, which in turn can be used to compare humans with other primates. Such comparisons facilitate the formation of evolutionary hypotheses to account for the particularities of human behavior and cognition.This paper is a good example of the use of dynamical modeling to distill interpretable information from neuroimaging data. The results are qualitatively clear: the differences between humans and other primates are visible in the figures. The findings related to a gradient of dynamic ranges along the anterior-posterior axis are also of wide interest and can be readily compared with studies on anatomical and physiological gradients. The study involves validation using a second computational model, enhancing the robustness of the results. The connection with computational capacity is also very interesting. Further, the model is used to make specific predictions about functional connectivity, which are verified. The results will be of interest to several subdisciplines within neuroscience: in particular, the link between connectivity and computational capacity has the potential to dovetail with more fine-grained types of data collected in other species, as well as with computational models of decision-making and evidence-accumulation.

We thank the Reviewer for their feedback and appreciation of our work.

The primary weakness of the paper lies in the difficulty of interpreting the broader meaning of the results. Readers may not readily understand what the behavioral and psychological consequences of high and low dynamic ranges are. Some further elaboration of this concept, perhaps with examples from perception and decision-making, will increase the potential readership. Additional explanation of why responses to changes in global recurrent strength are relevant to the local dynamic ranges of each cortical region/network will also be helpful.

In line with the Reviewer’s comment, we have amended the Results section to better explain the meaning of dynamic range. Moreover, we have expanded our explanation for potential mechanisms that can affect local excitability. See the following new text added to the Results section:

Lines 111-118: “In particular, previous work has shown that neuromodulatory agents can modify the biophysical properties of neurons through various cellular mechanisms [21]. One mechanism is via activation of metabotropic receptors that bring the resting membrane potential of neurons closer to their firing threshold [22]. This mechanism can mediate changes in the excitability of brain regions at the subsecond timescale [23], effectively driving modulations in the regional strength of recurrent connections (i.e., our model's w parameter). However, we clarify that the neuromodulation mechanism described above is only one example of many potential mechanisms that can drive changes in regional excitability.”

Lines 119-127: “We characterize the shape of the response function (i.e., the slope) demonstrated in Figure 2C in terms of the *neural dynamic range*, such that high dynamic range means that a region can respond to a wide range of changes in excitability (w), albeit the transition between activity levels is slow (red curve in the top panel of Figure 2D). Conversely, regions with a low dynamic range can quickly transition to high levels of activity with small changes in excitability, specifically at a critical intermediate regime (blue curve in the top panel of Figure 2D). When the dynamic range is very close to zero, the response function in Figure 2C becomes like a step function with infinite slope; hence, the response jumps between low and high activity levels, analogous to a phase transition.”

Lines 130-137: “Brain regions with similarly low or high dynamic ranges are more likely to reach equivalent functional states. This can be observed in the example time series at the bottom panel of Figure 2D, where regions with similarly low dynamic ranges have activity amplitudes fluctuating at similar levels across varying excitability regimes. Note that similar observations occur for regions with similarly high dynamic ranges. Moreover, regions with similar dynamic ranges have higher levels of correlated activity compared to regions with different dynamic ranges, suggesting better integration (e.g., see correlations of the time series in the bottom versus top panels of Figure 2D at corresponding w values).”

The interpretability issue also arises for the treatment of computational capacity: some elaboration of the differences and trade-offs when comparing accuracy in humans and chimpanzees will help with readability. The concept of "decision accuracy of the whole brain" is also somewhat obscure: it is not obvious why a decision should be construed as the result of independent drift-diffusion processes. It may also be helpful to comment briefly on how such a framework can be extended when a decision process is not a two-alternative forced choice.

Throughout the text, we have been careful not to make overarching claims about human and chimpanzee cognitive abilities as we do not have behavioral data to support them. Hence, we provided insights into the neural bases of behavior by assessing the computational capacity of human and chimpanzee connectomes. Specifically, we have assessed the capacity of each connectome to accurately reach a decision by calculating the average decision accuracy of all regions. However, we emphasize that even though the regions have separate drift-diffusion equations to govern their dynamics, they are in principle not independent as we include an inter-regional coupling term that takes into account region-to-region interactions scaled by their structural connectivity weights (Figure 6B). Nonetheless, we acknowledge that decision-making tasks may only recruit a set of brain regions as opposed to all brain regions. Our work provides a framework for future computational work aiming to assess how the brain’s wiring impacts decision-making processes.

We have added the following text to the Results section to better clarify our approach:

Lines 266-267: “Whole-brain accuracy represents the average of the accuracy of all regions.”

Lines 290-293: “Note that we adopt a generalized definition of decision accuracy based on the performance of the connectomes. Specifically, we do not take into account the possibility that only a subset of brain regions are recruited in the decision-making process.”

In line with the above, recent studies have shown that it is possible for the drift-diffusion model to be mathematically extended to multiple-alternative forced-choice decision making (Roxin, 2019, Journal of Mathematical Neuroscience). Hence, our framework can be applied to these extensions of the drift-diffusion model to investigate, for example, competition between regions. We have added the following text to the Discussion section to highlight possible extensions of our work:

Lines 435-438: “Moreover, our work provides a framework to investigate the relationship between whole-brain connectivity and neural dynamics across a wider family of species (e.g., across the mammalian species [63]) and its effects on complex cognitive processes (e.g., decision making with more than two alternatives [64]).”

The nature of the anatomical difference between humans and other primates is also somewhat unclear. The results are described as suggesting evidence for structural changes over the course of evolution. The difference in dynamic range distribution between humans and other primates serves as indirect evidence of these structural changes. But the results do not seem to include a direct comparison of the structural data (prior to modeling, using descriptive statistics) across the various primate species. In addition to this empirical question, the model suggests the possibility of generating specific hypotheses regarding the trajectory of human evolution. Is increased speed/myelination the key factor, or does the overall magnitude of connection weights matter too? What alterations of the chimpanzee connectivity dataset would bring the dynamic range distribution in line with that of humans? The information required to suggest answers to such questions seems to be in the paper, but it is not easy for the reader to piece it together.

Similar to Response 1.5, we have added new analyses and figures to better illustrate the structural and topological similarities and differences in the human and chimpanzee connectomes. In particular, we have added a new Results section “Human and chimpanzee connectomes”. In this section, we have included new analyses of connections specific to and shared between humans and chimpanzees. We have also provided a new graph theoretical analysis to compare the topological organization of the human and chimpanzee connectomes, providing further insights as to what the chimpanzee connectome would need in order to have similar distributions of dynamic ranges as the human connectome. We note that our results highlight that differences between the human and chimpanzee connectomes emerge from the interaction of various characteristics (e.g., connection weights and conduction speed) and dynamic range. Unpacking the distinct contribution of each factor can be pursued in follow-up studies.

We have added the following text to the Results and Materials and methods sections and have added a new Figure 1:

Lines 62-79: “We then normalize the group-averaged connectomes with respect to their maximum weights. Using the resulting connectomes, we examine connections present in one species but absent in the other (labeled as human-specific and chimpanzee-specific connections; Figure 1C). We note that the use of the term “specific” does not necessarily imply that said connections are unique to each of the species; i.e., they are only specific based on comparison of the connectivity strength of connections between the two species in our dataset. We find that intrahemispheric pathways comprise 82.6% (19 out of 23) of human-specific connections and 50% (3 out of 6) of chimpanzee-specific connections, a finding consistent with previous comparative connectome investigations [7,10]. We also examine the set of connections that are present in both species, termed shared connections (Figure 1C), and confirm that there is a strong correlation between connectivity strengths across both species (Figure 1D), consistent with previous studies [7,10]. At the whole-brain level, the human and chimpanzee connectomes largely overlap in their topological organization. In particular, the connectomes show similar levels of small-worldness (small-world propensity [11] values of 0.83 and 0.84 in human and chimpanzee, respectively) and modularity (modularity values of 0.54 and 0.56 in human and chimpanzee, respectively) [12,13]. At the regional level, the connectomes exhibit similar distributions of clustering coefficients (Figure 1E). On the other hand, human brain regions have significantly shorter path lengths compared to chimpanzee brain regions (Figure 1F).”

Lines 535-547: “To investigate the structural property of the connectomes, we leveraged concepts from the field of graph theory [11,13,74,76]. In particular, we quantified small-world propensity, modularity, regional clustering coefficient, and regional path length. Small-world propensity quantifies the extent to which the network exhibits a small-world structure [11]. Modularity quantifies the extent to which the network may be subdivided into distinct modules (i.e., groups of regions). The clustering coefficient quantifies the probability of finding a connection between the neighbors of a given node (region). Specifically, this metric was estimated by calculating the fraction of triangles around a region. The path length quantifies the level of integration in the network (short path length implying high integration). Path length corresponds to the total topological distance of the shortest path between two regions. For our weighted connectomes, we defined the topological distance to be inversely proportional to the weight of connection (i.e., distance = 1/weight). For each brain region, the regional path length was calculated by taking the average of the path lengths between that region and all other regions.”

Given that the methods come at the end, some comments on terminology would be helpful in the introduction and/or the results. It is not clear why the term "gating" is used instead of "output" or "response". A reader without a computational background may not be familiar with this term.

In the original formulation of the neural model used in the study, the variable *S* represents the fraction of open NMDA channels; hence, the use of the term gating.

To better guide the readers, we have replaced all instances of “synaptic gating” with “synaptic response” throughout the text and figures. We have also added the following text to the Results and Materials and methods sections to better explain the variable S:

Lines 98-99: “The variable S represents the fraction of activated NMDA channels; hence, higher S values correspond to higher neural activity and firing rates.”

Lines 557-559: “In the original formulation of the model, the synaptic response variable Si in Equation (1) acts as a gating variable that represents the fraction of open (or activated) NMDA channels [14–16]. Thus, higher values of Si correspond to higher neural activity.”

Why would neuromodulation only affect recurrent strength and not the off-diagonal terms of the connection matrix (matrix A)?

It is true that neuromodulation encompasses a variety of mechanisms, which could have both local and global effects in the brain. We anchored our discussion on the former effect based on recent studies showing the ability of neuromodulation to modify the local biophysical properties of neurons within a region (see review by Shine et al., 2021, Nature Neuroscience). For example, neuromodulation can activate metabotropic receptors, leading to the release of intracellular ca^2+^ and an increase in resting transmembrane potential. This process effectively changes the excitability of neurons within regions, which is captured by our recurrent strength parameter *w*.

We have added the following text to the Results section to explain the abovementioned potential neuromodulatory mechanisms affecting local dynamics:

Lines 111-118: “In particular, previous work has shown that neuromodulatory agents can modify the biophysical properties of neurons through various cellular mechanisms [21]. One mechanism is via activation of metabotropic receptors that bring the resting membrane potential of neurons closer to their firing threshold [22]. This mechanism can mediate changes in the excitability of brain regions at the subsecond timescale [23], effectively driving modulations in the regional strength of recurrent connections (i.e., our model's w parameter). However, we clarify that the neuromodulation mechanism described above is only one example of many potential mechanisms that can drive changes in regional excitability.”

The drift-diffusion model may not be sufficient to account for certain aspects of decision-making, such as urgency (e.g., Carland et al., 2015), or direct inhibition-mediated competition between alternatives. The differences in speed between humans and chimpanzees could conceivably point to differences in the amount and/or timescale of inhibition. Inhibition seems to be higher in humans than in other mammals: this may be relevant to the slower integration of evidence. Even in a model that lumps excitation and inhibition together as positive and negative connection weights, it may be illuminating to know if the E/I ratio sheds light on the speed of arriving at the decision threshold. This is not essential, but inhibition-related inferences would greatly enhance the interest of the paper.

We thank the Reviewer for this insightful comment. We have now added a new analysis to test whether the amount of inhibition could explain the difference in the performance of human and chimpanzee connectomes at earlier time periods. Following works by Carland et al., (2015; Psychonomic Bulletin and Review) and Lam et al., (2022, Journal of Neuroscience), we have extended our original drift-diffusion model to include a linear self-coupling term λy in Equation (15), where λ scales the self-coupling term and quantifies the acceleration or deceleration of the accumulation of decision according to its sign (see new Figure 6—figure supplement 3A). In particular, λ>0 corresponds to increased excitation and λ<0 corresponds to increased inhibition. Therefore, one can vary the λ parameter to test the effects of excitation/inhibition on the speed and accuracy of decision making. We found that more inhibition (i.e., λ<0) leads to slower decision times (i.e., time it takes for a region to reach a decision threshold) across all regions compared to the original λ=0 case (Figure 6—figure supplement 3B). Conversely, more excitation (i.e., λ>0) leads to faster decision times, with the caveat of poorer decision accuracies (Figure 6—figure supplement 3C). When both human and chimpanzee connectomes have the same level of excitation or inhibition, we were able to replicate our original results. However, more inhibition extends periods of inferior whole-brain decision accuracy in humans compared to the chimpanzee brain at earlier times.

Moreover, we found that for the decision accuracy of the human brain to be at par with that of the chimpanzee brain at earlier times (i.e., at tmin in Figure 6—figure supplement 3D), the human brain required extra excitation (i.e., λ=2.03) (Figure 6—figure supplement 3E). This result suggests that our original finding in Figure 6E is driven not only by the heterogeneous dynamic ranges of chimpanzee brain regions but also by higher intrinsic levels of inhibition in the human brain compared to the chimpanzee brain. This is certainly something that could be further explored in future works.

We have added the following text to the Results, Discussion, and Materials and methods sections and have added a new Figure 6—figure supplement 3:

Lines 299-311: “We next investigate whether levels of excitation and inhibition could have also influenced the difference between the decision accuracy of human and chimpanzee brains at earlier times (Figure 6E). Hence, we extend the drift-diffusion model in Figure 6B by incorporating a self-coupling term parametrized by λ (Figure 6—figure supplement 3A; λ>0 and λ<0 corresponds to increased excitation and inhibition, respectively) [36,37]. We find that increased excitation leads to faster decision times (Figure 6—figure supplement 3B) but poorer overall decision accuracy (Figure 6—figure supplement 3C). We also find that increased inhibition extends periods of inferior whole-brain decision accuracy of the human brain compared to the chimpanzee brain at earlier times (Figure 6—figure supplement 3D). Interestingly, we also find that the human brain requires additional level of excitation (i.e., λ=2.03) at earlier times in order to reach the level of decision accuracy achieved by the chimpanzee brain (Figure 6—figure supplement 3E). This result suggests that our original finding in Figure 6E could also be driven by higher intrinsic levels of inhibition in the human brain.”

Lines 410-414: “This latter result was attributed to the (i) more heterogeneous distribution of dynamic ranges in chimpanzees, supporting that diversity in local neural properties is important for rapid computations [43], and (ii) higher intrinsic levels of inhibition in the human brain [36,37]. This regional diversity and higher intrinsic inhibition …”

Lines 634-636: “We also set λ=0 in our main results in Figure 6, but also extensively varied it in Figure 6—figure supplement 3 to investigate the effects of excitation (i.e., λ>0) and inhibition (i.e., λ<0) on the decision-making capacity of the connectomes.”

Using the term "diffuse" for high dynamic range seems a bit confusing without further elaboration.

To avoid confusion, we have changed the terms “diffuse” and “sharp” to “high” and “low”, respectively, throughout the text and figures.

Figure 1. A qualitative explanation of the sigmoidal shape would be useful. How can we interpret the two knee points of the sigmoid, as well as the slope? Is a very steep sigmoid (or a step function) equivalent to an all-or-nothing response to the crossing of a threshold in recurrent activity?

The Reviewer is correct that the dynamic range effectively captures the slope of the sigmoidal shape of the response function. Hence, the closer the dynamic range is to zero the more the response function looks like a step function. This means that the response exhibits a jump transition between low and high levels of activity.

We have added the following text to the Results section to provide a clearer interpretation of the response function and dynamic range:

Lines 119-127: “We characterize the shape of the response function (i.e., the slope) demonstrated in Figure 2C in terms of the *neural dynamic range*, such that high dynamic range means that a region can respond to a wide range of changes in excitability (w), albeit the transition between activity levels is slow (red curve in the top panel of Figure 2D). Conversely, regions with a low dynamic range can quickly transition to high levels of activity with small changes in excitability, specifically at a critical intermediate regime (blue curve in the top panel of Figure 2D). When the dynamic range is very close to zero, the response function in Figure 2C becomes like a step function with infinite slope; hence, the response jumps between low and high activity levels, analogous to a phase transition.”

Page 1 Line 13: Some examples of topological properties would be useful.

We have added example topological properties found to be conserved across species. We have revised the following lines in the Introduction accordingly:

Lines 43-44: “ topological properties (e.g., small-world and modularity properties [7])”

Page 3 line 10: "Brain regions with similar dynamic ranges are more likely to coactivate, allowing efficient region-to-region integration of neural processes." Why are they more likely to coactivate? Is there a probabilistic argument? One might assume instead that brain regions with high/diffuse dynamic ranges would be more likely to coactivate, since there are more opportunities for them to become active in the first place.

The statement highlighted by the Reviewer was originally intended to mean that regions with similar dynamic ranges (regardless whether they are high or low) have similar dynamical properties and are more likely to reach equivalent functional states, hence the use of the term “coactivate”. This is visually shown in the new Figure 2D, where we provide example time series of regions with different dynamic ranges when the global recurrent strength w is increased by about 30%. We see that when regions have different dynamic ranges (top panel of Figure 2D), the resulting amplitudes of synaptic responses fluctuate at different levels and the time series at both *w* regimes have low correlations (0.06 and 0.08 at w = 0.6 and w = 0.8, respectively). However, when regions have similar dynamic ranges (bottom panel of Figure 2D), the magnitudes fluctuate at similar levels and the time series have higher correlations (0.40 and 0.14 at w = 0.6 and w = 0.8, respectively). Whilst the example at the bottom panel of Figure 2D only shows the low dynamic range case, similar observations exist in the high dynamic range case.

To avoid confusion, we have removed the term “coactivate” throughout the text. We have also revised the relevant line in the Results section highlighted by the Reviewer and have added a new Figure 2D:

Lines 130-137: “Brain regions with similarly low or high dynamic ranges are more likely to reach equivalent functional states. This can be observed in the example time series at the bottom panel of Figure 2D, where regions with similarly low dynamic ranges have activity amplitudes fluctuating at similar levels across varying excitability regimes. Note that similar observations occur for regions with similarly high dynamic ranges. Moreover, regions with similar dynamic ranges have higher levels of correlated activity compared to regions with different dynamic ranges, suggesting better integration (e.g., see correlations of the time series in the bottom versus top panels of Figure 2D at corresponding w values).”

Line 398: “… are more likely to reach equivalent functional states …”